# Dynamic Regret of Adversarial Linear Mixture MDPs

**Long-Fei Li, Peng Zhao, Zhi-Hua Zhou**

National Key Laboratory for Novel Software Technology, Nanjing University, China

School of Artificial Intelligence, Nanjing University, China

{lilf, zhaop, zhouzh}@lamda.nju.edu.cn

## Abstract

We study reinforcement learning in episodic inhomogeneous MDPs with adversarial full-information rewards and the unknown transition kernel. We consider the linear mixture MDPs whose transition kernel is a linear mixture model and choose the *dynamic regret* as the performance measure. Denote by $d$ the dimension of the feature mapping, $H$ the length of each episode, $K$ the number of episodes, $P_T$ the non-stationary measure, we propose a novel algorithm that enjoys an $\widetilde{\mathcal{O}}\big(\sqrt{d^2 H^3 K} + \sqrt{H^4(K + P_T)(1 + P_T)}\big)$ dynamic regret under the condition that $P_T$ is known, which improves previously best-known dynamic regret for adversarial linear mixture MDP and adversarial tabular MDPs. We also establish an $\Omega\big(\sqrt{d^2 H^3 K} + \sqrt{HK(H + P_T)}\big)$ lower bound, indicating our algorithm is *optimal* in $K$ and $P_T$. Furthermore, when the non-stationary measure $P_T$ is unknown, we design an online ensemble algorithm with a meta-base structure, which is proved to achieve an $\widetilde{\mathcal{O}}\big(\sqrt{d^2 H^3 K} + \sqrt{H^4(K + P_T)(1 + P_T) + H^2 S_T^2}\big)$ dynamic regret and here $S_T$ is the expected switching number of the best base-learner. The result can be optimal under certain regimes.

## 1 Introduction

Reinforcement Learning (RL) aims to learn a policy that maximizes the cumulative reward through interacting with the environment, which has achieved tremendous successes in various fields, including games [1, 2], robotic control [3, 4], and dialogue generation [5, 6]. In reinforcement learning, the Markov Decision Process (MDP) [7] is the most widely used model to describe the environment.

Traditional MDPs assume that the reward functions are stochastic and the number of actions and states is small. However, in many real-world applications, the reward functions may be adversarially changing and the state and action spaces are large or even infinite. Previous work studies these two problems separately. To deal with the adversarial reward functions, Even-Dar et al. [8] first consider learning *adversarial* MDPs with known transition and full-information reward feedback. They propose the MDP-E algorithm that enjoys $\widetilde{\mathcal{O}}(\sqrt{\tau^3 T})$ regret, where $\tau$ is the mixing time and $T$ is the number of steps. There is a line of subsequent work studying adversarial MDPs [9, 10, 11, 12, 13, 14, 15, 16], which studies various settings depending on whether the transition kernel is known, and whether the feedback is full-information or bandit. To overcome the large state and action space issue, a widely used method is *linear function approximation*, which reparameterizes the value function as a linear function over some feature mapping that maps the state and action to a low-dimensional space. Amongst these work, linear MDPs [17, 18, 19, 20, 21, 22, 23] and linear mixture MDPs [24, 25, 26, 27, 28, 29, 30, 31] are two of the most popular MDP models with linear function approximation. In particular, He et al. [23] and Zhou et al. [31] attain the minimax optimal $\widetilde{\mathcal{O}}(\sqrt{d^2 H^3 K})$ regret for linear MDPs and linear mixture MDPs with stochastic rewards respectively.

Recent studies try to combine two lines of work to establish the theoretical foundation of adversarial MDPs with large state and action space. In particular, Cai et al. [32] study adversarial linear mixture

37th Conference on Neural Information Processing Systems (NeurIPS 2023).

MDPs and propose the OPPO algorithm that enjoys an $\widetilde{\mathcal{O}}(\sqrt{d^2 H^4 K})$ regret. He et al. [33] improve the result to $\widetilde{\mathcal{O}}(\sqrt{d^2 H^3 K})$ and show it is (nearly) optimal by presenting a matching lower bound. However, these studies choose *static regret* as the performance measure, defined as the performance difference between the learner's policy and that of the best-fixed policy in hindsight, namely,

$$\text{Regret}(K) = \max_{\pi \in \Pi} \sum_{k=1}^{K} V_{k,1}^{\pi}(s_{k,1}) - \sum_{k=1}^{K} V_{k,1}^{\pi_k}(s_{k,1}), \tag{1}$$

where $V$ is the value function and $\Pi$ is the set of all stochastic policies. One caveat in (1) is that the best *fixed* policy may behave poorly in non-stationary environments. To this end, we introduce *dynamic regret*, which benchmarks the learner's performance with changing policies, defined as

$$\text{D-Regret}(K) = \sum_{k=1}^{K} V_{k,1}^{\pi_k^c}(s_{k,1}) - \sum_{k=1}^{K} V_{k,1}^{\pi_k}(s_{k,1}), \tag{2}$$

where $\pi_1^c, \ldots, \pi_K^c \in \Pi$ is compared policies. Define $P_T = \sum_{k=2}^{K} d(\pi_k^c, \pi_{k-1}^c)$ with a certain distance measure $d(\cdot, \cdot)$ as the non-stationary measure. A favorable dynamic regret should scale with $P_T$.

Dynamic regret is a more appropriate measure in non-stationary environments, but it is more challenging to optimize such that few studies focus on it in the literature. Zhao et al. [34] investigate the dynamic regret of adversarial tabular MDPs with the *known* transition kernel and present an algorithm with optimal dynamic regret. Importantly, their algorithm does not require the non-stationarity measure $P_T$ as the algorithmic input ahead of time. For the unknown transition setting, Fei et al. [35] study adversarial tabular MDPs and propose an algorithm with dynamic regret guarantee. Zhong et al. [36] further extend the algorithm of Fei et al. [35] to accommodate non-stationary transition kernels with linear function approximation. Both algorithms of Fei et al. [35] and Zhong et al. [36] require the quantity of $P_T$ as the input. Moreover, their dynamic regret bounds are suboptimal in terms of $K$ and $P_T$ as demonstrated by the lower bound established by our work (see Theorem 4).

This work investigates the dynamic regret of adversarial linear mixture MDPs, with a focus on the full-information feedback and the unknown transition. We first propose POWERS-FixShare algorithm when $P_T$ is known, an algorithm combining optimistic policy optimization with a Bernstein bonus and fixed-share mechanism. We show it enjoys an $\widetilde{\mathcal{O}}(\sqrt{d^2 H^3 K} + \sqrt{H^4 (K + P_T)(1 + P_T)})$ dynamic regret, where $d$ is the dimension of the feature mapping, $H$ is the length of each episode, $K$ is the number of episodes, $P_T$ is the non-stationary measure. We also establish a dynamic regret lower bound of $\Omega(\sqrt{d^2 H^3 K} + \sqrt{HK(H + P_T)})$. We stress four remarks regarding our results:

(1) Our result can recover the $\widetilde{\mathcal{O}}(\sqrt{d^2 H^3 K})$ minimax optimal static regret in He et al. [33].

(2) Our result improves upon the previously best-known $\widetilde{\mathcal{O}}(dH^{7/4} K^{3/4} + H^2 K^{2/3} P_T^{1/3})$ dynamic regret for the fixed transition of Zhong et al. [36, Theorem 4.6] in terms of $H$, $K$ and $P_T$.

(3) Our result can imply an $\widetilde{\mathcal{O}}(\sqrt{H^4 S^2 A K} + H^2 \sqrt{(K + P_T)(1 + P_T)})$ dynamic regret for adversarial tabular MDPs, strictly improving upon the previously best-known $\widetilde{\mathcal{O}}(\sqrt{H^4 S^2 A K} + H^2 K^{2/3} P_T^{1/3})$ dynamic regret of Fei et al. [35, Theorem 1], in terms of both $K$ and $P_T$.

(4) As the lower bound suggests, our result is the first optimal regarding the dependence on $d$, $K$ and $P_T$ and can be optimal in terms of $H$ under certain regimes ($H \leq d$ and $P_T \leq d^2/H$).

Furthermore, we study the case when $P_T$ is unknown and design a novel algorithm equipped with the dynamic regret guarantee by the meta-base two-layer structure. Our algorithm enjoys an $\widetilde{\mathcal{O}}(\sqrt{d^2 H^3 K} + \sqrt{H^4 (K + P_T)(1 + P_T) + H^2 S_T^2})$ dynamic regret, where $S_T$ is the expected switching number of the best base-learner. Though $S_T$ is a data-dependent quantity, it also reflects the degree of environmental non-stationarity to some extent. Moreover, under specific regimes, the magnitude of $S_T$ may be relatively negligible, resulting in our results still being optimal. Indeed, given that $S_T$ is a data-dependent quantity, its inclusion in the regret bound is not ideal. Deriving bounds that rely exclusively on problem-dependent quantities, like $P_T$, remains an open challenge. We discuss the technical difficulty of removing $S_T$ in Section 5 and take this issue for future work.

Finally, we also highlight the main technical challenges and our solutions as follows.

- We first show that the dynamic regret depends on the inverse of the minimal probability over the action space of our policies, which can be arbitrarily small. To this end, we propose a novel

algorithm with the *fixed-share* mechanism [37]. While this mechanism is proved to enjoy favorable dynamic regret in online convex optimization [38], it suffers an additional term that can be regarded as the weighted sum of the difference between the occupancy measure of the compared policies in online MDPs. To overcome the difficulty, we exploit the *multiplicative stability* to bound this term, eliminating the need for a restart strategy to handle the environmental non-stationarity as in previous studies [35, 36] and allows us to attain the dynamic regret optimal in terms of $K$ and $P_T$.

- We show the dynamic regret of online MDPs can be written as the weighted average of "multi-armed bandits" problems over all states, where the weight for each state is the *unknown* and *changing* probability of being visited by $\pi_1^c, \ldots, \pi_K^c$. For the unknown $P_T$ case, we first show the standard two-layer structure used in non-stationary online learning studies [39, 40, 41] fails to achieve a favorable dynamic regret guarantee, which characterizes the unique difficulty of online MDPs. Then, we present an initial attempt to address this issue by a specifically designed two-layer structure. We prove our algorithm enjoys nice dynamic regret guarantees under certain regimes.

**Notations.** We denote by $\Delta(\mathcal{A})$ the set of probability distributions on a set $\mathcal{A}$ and denote the KL-divergence by $D_{\mathrm{KL}}(p\|p') = \sum_{a \in \mathcal{A}} p(a) \log \frac{p(a)}{p'(a)}$ for any $p, p' \in \Delta(\mathcal{A})$. We define $\Delta(\mathcal{A} \mid \mathcal{S}, H) = \{\{\pi_h(\cdot \mid \cdot)\}_{h=1}^H \mid \pi_h(\cdot \mid x) \in \Delta(\mathcal{A}), \forall s \in \mathcal{S}, h \in [H]\}$ for any set $\mathcal{S}$ and $H \in \mathbb{Z}_+$. Further, for any $\pi, \pi', \pi'' \in \Delta(\mathcal{A} \mid \mathcal{S}, H)$, we define $\mathbb{E}_\pi[\widetilde{D}_{\mathrm{KL}}(\pi'\|\pi'')] = \mathbb{E}_\pi[\sum_{h=1}^H D_{\mathrm{KL}}(\pi_h'(\cdot \mid s_h)\|\pi_h''(\cdot \mid s_h))]$. For any policy pair $\pi_h, \pi_h'$, we define $\|\pi_h - \pi_h'\|_{1,\infty} = \max_{s \in \mathcal{S}} \|\pi_h(\cdot \mid s) - \pi_h'(\cdot \mid s)\|_1$. For any $a, b, x \in \mathbb{R}$ with $a \le b$, let $[x]_{[a,b]}$ denote $\min\{\max(x, a), b\}$. $\widetilde{\mathcal{O}}(\cdot)$ omits the logarithmic factors.

## 2 Related Work

**RL with adversarial rewards.** There are many studies on learning adversarial MDPs where the reward functions are adversarially chosen, yielding fruitful results that can be categorized into three lines [8, 9, 10, 11, 12, 13, 14, 15, 16]. In particular, the first line of work considers the infinite-horizon MDPs with uniform mixing assumption. In the known transition and full-information setting, the seminal work of Even-Dar et al. [8] proposes MDP-E algorithm that achieves the $\widetilde{\mathcal{O}}(\sqrt{\tau^3 T})$ regret, where $\tau$ is the mixing time and $T$ is the number of steps. Another concurrent work of Yu et al. [9] achieves $\widetilde{\mathcal{O}}(T^{2/3})$ in the same setting. In the known transition and bandit-feedback setting, Neu et al. [10] propose MDP-EXP3 algorithm that attains $\widetilde{\mathcal{O}}(T^{2/3})$ regret. The second line of work considers the episodic loop-free MDPs. Neu et al. [11] first study this problem under the known transition setting and propose algorithms that achieve $\widetilde{\mathcal{O}}(\sqrt{T})$ and $\widetilde{\mathcal{O}}(\sqrt{T/\alpha})$ for full-information and bandit-feedback respectively, where $\alpha$ is the lower bound of the probability of all states under any policy. Zimin and Neu [12] propose O-REPS algorithm that enjoys $\widetilde{\mathcal{O}}(\sqrt{T})$ regret in both full-information and bandit-feedback setting without any additional assumption. Rosenberg and Mansour [13] and Jin et al. [14] further consider the harder unknown transition and bandit-feedback setting. The last line of work studies the episodic Stochastic Shortest Path (SSP) setting [15, 16]. In this paper, we focus on episodic MDPs with the unknown transition and full-information setting.

**RL with linear function approximation.** To design RL algorithms in large state and action space scenarios, recent works focus on solving MDPs with linear function approximation. In general, these works can be divided into three lines based on the specific assumption of the underlying MDP. The first line of work considers the low Bellman-rank assumption [42, 43, 44, 45], which assumes the Bellman error matrix has a low-rank factorization. The second line of work is based on the linear MDP assumption [17, 18, 19, 20, 21, 22, 23], where both the transition kernel and reward functions can be parameterized as linear functions of given state-action feature mappings $\phi : \mathcal{S} \times \mathcal{A} \to \mathbb{R}^d$. The last line of work studies the linear mixture MDP [24, 25, 26, 28, 29, 30, 31], where the transition kernel can be parameterized as a linear function of a feature mapping $\phi : \mathcal{S} \times \mathcal{A} \times \mathcal{S} \to \mathbb{R}^d$ but without the linear reward functions assumption. Amongst these works, He et al. [23] and Zhou et al. [31] attain the minimax optimal $\widetilde{\mathcal{O}}(\sqrt{d^2 H^3 K})$ regret for both episodic linear MDPs and linear mixture MDPs respectively. However, all the above studies consider the stochastic reward setting. In this paper, we study the episodic linear mixture MDP setting but with adversarial reward functions.

**Non-stationary RL.** Another related line of research is online non-stationary MDPs. In contrast to adversarial MDPs where the reward functions are generated in an adversarial manner, online non-stationary MDPs consider the setting where the reward functions are generated stochastically

according to some distributions that may vary over time. Jaksch et al. [46] study the piecewise stationary setting where the transitions and rewards are allowed to change certain times and propose UCRL2 with restart technique to deal with the non-stationarity. Later, Ortner et al. [47] consider the generalized setting where the changes are allowed to take place every step. However, the above studies need prior knowledge about the magnitude of non-stationary. To address this issue, Cheung et al. [48] propose the Bandit-over-RL algorithm to remove this requirement. A recent breakthrough by Wei and Luo [49] introduces a black-box method that can convert any algorithm satisfying specific conditions and enjoying optimal static regret in stationary environments into another with optimal dynamic regret in non-stationary environments, without requiring prior knowledge about the degree of non-stationarity. However, this approach does not apply to the adversarial setting. Specifically, their reduction requires the base algorithm to satisfy a certain property enjoyed by typical UCB-type algorithms. When a new instance of the base algorithm surpasses the optimistic estimator, it can be inferred that the environment has changed, prompting a restart of the algorithm to disregard prior information. However, this approach of constructing an optimistic estimator by a UCB-type algorithm can only be applied to a *stochastic* setting. In the *adversarial* setting, where no model assumptions are made and comparators can be arbitrary, this approach encounters significant difficulties.

**Dynamic Regret.** Dynamic regret of RL with adversarial rewards is only recently studied in the literature [34, 35, 36]. Zhao et al. [34] investigate the dynamic regret of adversarial tabular MDPs with the *known* transition kernel and present an algorithm with optimal dynamic regret. Importantly, their algorithm does not require the non-stationarity measure $P_T$ as the algorithmic input ahead of time. For the unknown transition setting, Fei et al. [35] study adversarial tabular MDPs and propose an algorithm with dynamic regret guarantees. Zhong et al. [36] further extend the algorithm of Fei et al. [35] to accommodate non-stationary transition kernels with linear function approximation. Both algorithms [35, 36] require the quantity of $P_T$ as the input. Moreover, their dynamic regret bounds are suboptimal in $K$ and $P_T$ as shown by the lower bound established in our work. In this work, we first design an *optimal* algorithm in terms of $K$ and $P_T$ when $P_T$ is known. Further, we develop the first algorithm to handle the unknown $P_T$ issue in adversarial MDPs with unknown transition.

## 3 Problem Setup

We focus on episodic inhomogeneous MDPs with full-information reward functions and the unknown transition kernel. Denote by $M = \{\mathcal{S}, \mathcal{A}, H, \{r_{k,h}\}_{k\in[K],h\in[H]}, \{\mathbb{P}_h\}_{h\in[H]}\}$ an episodic inhomogeneous MDP, where $\mathcal{S}$ is the state space, $\mathcal{A}$ is the action space, $K$ is the number of episodes, $H$ is the horizon, $r_{k,h} : \mathcal{S} \times \mathcal{A} \to [0,1]$ is the reward function, $\mathbb{P}_h(\cdot \mid \cdot, \cdot) : \mathcal{S} \times \mathcal{A} \times \mathcal{S} \to [0,1]$ is the transition kernel. We assume the rewards are deterministic without loss of generality and extending our results to stochastic rewards is straightforward. Let $S = |\mathcal{S}|$ and $A = |\mathcal{A}|$.

The learner interacts with the MDP through $K$ episodes without knowledge of transition kernel $\{\mathbb{P}_h\}_{h\in[H]}$. In each episode $k$, the environment chooses the reward function $\{r_{k,h}\}_{h\in[H]}$ and decides the initial state $s_{k,1}$, where the reward function may be chosen in an adversarial manner and depend on the history of the past $(k-1)$ episodes. Simultaneously, the learner decides a policy $\pi_k = \{\pi_{k,h}\}_{h\in[H]}$ where each $\pi_{k,h} : \mathcal{S} \to \Delta(\mathcal{A})$ is a function that maps a state $s$ to a distribution over action space $\mathcal{A}$. In the $h$ stage in episode $k$, the learner observes current state $s_{k,h}$, chooses an action $a_{k,h} \sim \pi_{k,h}(\cdot \mid s_{k,h})$, and transits to the next state $s_{k,h+1} \sim \mathbb{P}_h(\cdot \mid s_{k,h}, a_{k,h})$. Then the learner obtains the reward $r_{k,h}(s_{k,h}, a_{k,h})$ and observes the reward function $r_{k,h}$ as we consider the full-information setting. At stage $H+1$, the learner observes the final state $s_{k,H+1}$ but does not take any action, and the episode $k$ terminates. Denote by $T = KH$ the total steps throughout $K$ episodes.

**Linear Mixture MDPs.** In this work, we focus on a special class of MDPs called *linear mixture MDPs*, a setting initiated by Ayoub et al. [24] and further studied in the subsequent work [32, 31, 33]. In this setup, the transition kernel can be parameterized as a linear function of a feature mapping $\phi : \mathcal{S} \times \mathcal{A} \times \mathcal{S} \to \mathbb{R}^d$. The formal definition of linear mixture MDPs is as follows.

**Definition 1** (Linear Mixture MDPs). An MDP $M = \{\mathcal{S}, \mathcal{A}, H, \{r_{k,h}\}_{k\in[K],h\in[H]}, \{\mathbb{P}_h\}_{h\in[H]}\}$ is called an inhomogeneous, episode $B$-bounded linear mixture MDP, if there exist a *known* feature mapping $\phi(s' \mid s,a) : \mathcal{S} \times \mathcal{A} \times \mathcal{S} \to \mathbb{R}^d$ and an *unknown* vector $\theta_h^* \in \mathbb{R}^d$ with $\|\theta_h^*\|_2 \leq B$, $\forall h \in [H]$ such that (i) $\mathbb{P}_h(s' \mid s,a) = \phi(s' \mid s,a)^\top \theta_h^*$ for all $(s,a,s') \in \mathcal{S} \times \mathcal{A} \times \mathcal{S}$ and $h \in [H]$, (ii) $\|\phi_V(s,a)\|_2 \triangleq \|\sum_{s'\in\mathcal{S}} \phi(s' \mid s,a) V(s')\|_2 \leq 1$ for any $(s,a) \in \mathcal{S} \times \mathcal{A}$ and function $V : \mathcal{S} \to [0,1]$.

---

**Algorithm 1** POWERS-FixShare

---

**Input:** step size $\eta$, exploration parameter $\gamma$ and regularization parameter $\lambda$.

1: Initialize $\{\pi_{0,h}(\cdot\,|\,s)\}_{h=1}^H, s \in \mathcal{S}$ as uniform distribution, and set $\{Q_{0,h}(\cdot,\cdot)\}_{h=1}^H$ as zero function.

2: **for** $k = 1, 2, \cdots, K$ **do**

3:     Receive the initial state $s_{k,1}$.

4:     **for** $h = 1, 2, \cdots, H$ **do**

5:         For all $h \in [H], s \in \mathcal{S}$, updates policy by

$$\pi'_{k,h}(\cdot\,|\,s) \propto \pi_{k-1,h}(\cdot\,|\,s)\exp\left(\eta Q_{k-1,h}(s,\cdot)\right), \pi_{k,h}(\cdot\,|\,s) = (1-\gamma)\pi'_{k,h}(\cdot\,|\,s) + \gamma\pi^u(\cdot\,|\,s).$$

6:         Take the action following $a_{k,h} \sim \pi_{k,h}(\cdot\,|\,s_{k,h})$ and transit to the next state $s_{k,h+1}$.

7:         Obtain reward $r_{k,h}(s_{k,h}, a_{k,h})$ and observe the reward function $r_{k,h}(\cdot,\cdot)$.

8:     **end for**

9:     Initialize $V_{k,H+1}(\cdot)$ as a zero function.

10:     **for** $h = H, H-1, \cdots, 1$ **do**

11:         Set $Q_{k,h}(\cdot,\cdot) \leftarrow \left[r_{k,h}(\cdot,\cdot) + \langle\widehat{\theta}_{k,h}, \phi_{V_{k,h+1}}(\cdot,\cdot)\rangle + \widehat{\beta}_k\|\widehat{\Sigma}_{k,h}^{-1/2}\phi_{V_{k,h+1}}(\cdot,\cdot)\|_2\right]_{[0,H-h+1]}$.

12:         Set $V_{k,h}(\cdot) \leftarrow \mathbb{E}_{a\sim\pi_{k,h}(\cdot\,|\,\cdot)}[Q_{k,h}(\cdot,\cdot)]$.

13:         Set the estimated variance $[\bar{\mathbb{V}}_{k,h}V_{k,h+1}](s_{k,h}, a_{k,h})$ as in (11), bonus $E_{k,h}$ as in (21).

14:         $\bar{\sigma}_{k,h} \leftarrow \sqrt{\max\left\{H^2/d, [\bar{\mathbb{V}}_{k,h}V_{k,h+1}](s_{k,h}, a_{k,h}) + E_{k,h}\right\}}$.

15:         $\widehat{\Sigma}_{k+1,h} \leftarrow \widehat{\Sigma}_{k,h} + \bar{\sigma}_{k,h}^{-2}\phi_{V_{k,h+1}}(s_{k,h}, a_{k,h})\phi_{V_{k,h+1}}(s_{k,h}, a_{k,h})^\top$.

16:         $\widehat{b}_{k+1,h} \leftarrow \widehat{b}_{k,h} + \bar{\sigma}_{k,h}^{-2}\phi_{V_{k,h+1}}(s_{k,h}, a_{k,h})V_{k,h+1}(s_{h+1}^k)$.

17:         $\widetilde{\Sigma}_{k+1,h} \leftarrow \widetilde{\Sigma}_{k,h} + \phi_{V_{k,h+1}^2}(s_{k,h}, a_{k,h})\phi_{V_{k,h+1}^2}(s_{k,h}, a_{k,h})^\top$.

18:         $\widetilde{b}_{k+1,h} \leftarrow \widetilde{b}_{k,h} + \phi_{V_{k,h+1}^2}(s_{k,h}, a_{k,h})V_{k,h+1}^2(s_{h+1}^k)$.

19:         $\widehat{\theta}_{k+1,h} \leftarrow \widehat{\Sigma}_{k+1,h}^{-1}\widehat{b}_{k+1,h}, \widetilde{\theta}_{k+1,h} \leftarrow \widetilde{\Sigma}_{k+1,h}^{-1}\widetilde{b}_{k+1,h}$.

20:     **end for**

21: **end for**

---

**Dynamic Regret.** For any policy $\pi = \{\pi_h\}_{h\in[H]}$ and any $(k, h, s, a) \in [K] \times [H] \times \mathcal{S} \times \mathcal{A}$, we define the action-value function $Q_{k,h}^\pi$ and value function $V_{k,h}^\pi$ as

$$Q_{k,h}^\pi(s, a) = \mathbb{E}_\pi\left[\sum_{h'=h}^H r_{k,h'}(s_{h'}, a_{h'})\,\Big|\,s_h = s, a_h = a\right], V_{k,h}^\pi(s) = \mathbb{E}_\pi\left[\sum_{h'=h}^H r_{k,h'}(s_{h'}, a_{h'})\,\Big|\,s_h = s\right].$$

The Bellman equation is given by $Q_{k,h}^\pi = r_{k,h} + \mathbb{P}_h V_{k,h+1}^\pi$, and $V_{k,h}^\pi(s) = \mathbb{E}_{a\sim\pi_h(\cdot\,|\,s)}[Q_{k,h}^\pi(s, a)]$ with $V_{k,H+1}^\pi = 0$. For simplicity, for any function $V : \mathcal{S} \to \mathbb{R}$, we define the operator

$$[\mathbb{P}_h V](s, a) = \mathbb{E}_{s'\sim\mathbb{P}_h(\cdot\,|\,s,a)}V(s'), \quad [\mathbb{V}_h V](s, a) = [\mathbb{P}_h V^2](s, a) - ([\mathbb{P}_h V](s, a))^2. \tag{3}$$

As stated in Section 1, dynamic regret is a more appropriate measure compared with static regret for the adversarial environments, which is defined in (2) and we rewrite it below for clarity:

$$\text{D-Regret}(K) = \sum_{k=1}^K V_{k,1}^{\pi_k^c}(s_{k,1}) - \sum_{k=1}^K V_{k,1}^{\pi_k}(s_{k,1}), \tag{4}$$

where $\pi_1^c, \ldots, \pi_K^c$ is any sequence of compared policies. We define $\pi_0^c = \pi_1^c$ to simplify the notation. The non-stationarity measure is defined as $P_T = \sum_{k=1}^K \sum_{h=1}^H \|\pi_{k,h}^c - \pi_{k-1,h}^c\|_{1,\infty}$.

## 4   Optimal Dynamic Regret with Known $P_T$

We present our proposed algorithm in Algorithm 1. Similar to previous studies, the algorithm consists of (i) policy improvement phase, and (ii) policy evaluation phase. We introduce the details below. In Sections 4.1, we first consider the case when the transition is *known* to highlight the challenges even under the ideal setting. Then, we extend the results to the *unknown* transition setting in Section 4.2.

## 4.1 Policy Improvement Phase

In the policy improvement phase, the algorithm updates $\pi_k$ based on $\pi_{k-1}$ using the proximal policy optimization (PPO) method [50]. Specifically, at episode $k$, we define the following linear function:

$$L_{k-1}(\pi) = V_{k,1}^{\pi_{k-1}}(s_{k,1}) + \mathbb{E}_{\pi_{k-1}} \left[ \sum_{h=1}^{H} \langle Q_{k-1,h}^{\pi_{k-1}}, \pi_h(\cdot \mid s_h) - \pi_{k-1,h}(\cdot \mid s_h) \rangle \; \middle| \; s_1 = s_{k,1} \right],$$

which is the first-order Taylor approximation of $V_{k-1,1}^{\pi}(s_{k,1})$ around $\pi_{k-1}$. Then, we update $\pi_k$ by

$$\pi_k = \underset{\pi \in \Delta(\mathcal{A} \mid \mathcal{S}, H)}{\arg\max} \; L_{k-1}(\pi) - \frac{1}{\eta} \mathbb{E}_{\pi_{k-1}} \left[ \sum_{h=1}^{H} D_{\mathrm{KL}} \big( \pi_h(\cdot \mid s_h) \| \pi_{k-1,h}(\cdot \mid s_h) \big) \right], \tag{5}$$

where $\eta > 0$ is the stepsize and the KL-divergence encourages $\pi_k$ to be close to $\pi_{k-1}$ so that $L_{k-1}(\pi)$ is a good approximation of $V_{k-1,1}^{\pi}(s_{k,1})$. The update rule in (5) takes the following closed form,

$$\pi_{k,h}(\cdot \mid s) \propto \pi_{k-1,h}(\cdot \mid s) \cdot \exp \big( \eta \cdot Q_{k-1,h}^{\pi_{k-1}}(s, \cdot) \big), \forall h \in [H], s \in \mathcal{S}. \tag{6}$$

We show the update rule in (6) ensures the following guarantee and the proof is in Appendix C.1.

**Lemma 1.** *The update rule in* (6) *ensures the following dynamic regret guarantee:*

$$\text{D-Regret}(K) \leq \frac{\eta K H^3}{2} + \frac{1}{\eta} \sum_{k=1}^{K} \mathbb{E}_{\pi_k^c} \left[ \widetilde{D}_{\mathrm{KL}} \left( \pi_k^c \| \pi_k \right) - \widetilde{D}_{\mathrm{KL}} \left( \pi_k^c \| \pi_{k+1} \right) \right]. \tag{7}$$

Note that the expectation in the last term in (7) is taken over $\pi_k^c$ which may *change* over episode $k$. For static regret, i.e., $\pi_1^c = \ldots = \pi_K^c = \pi^*$, we can control this term by a standard telescoping argument, which is not viable for dynamic regret analysis. Fei et al. [35] propose a restart strategy to handle this term. Specifically, they restart the algorithm every certain number of steps and decompose the above expectation into $\mathbb{E}_{\pi_k^c}[\cdot] = \mathbb{E}_{\pi_{k_0}^c}[\cdot] + \mathbb{E}_{\pi_k^c - \pi_{k_0}^c}[\cdot]$ where $k_0 < k$ is the episode in which restart takes place most recently before episode $k$. For the first expectation $\mathbb{E}_{\pi_{k_0}^c}[\cdot]$, they apply a customized telescoping argument to each period as the expectation is taken over the fixed policy. The second expectation $\mathbb{E}_{\pi_k^c - \pi_{k_0}^c}[\cdot]$ involves the difference $\pi_k^c - \pi_{k_0}^c$ and can be bounded by $P_T$. However, as we will show in Theorem 4, their regret bound is suboptimal in terms of $K$ and $P_T$.

We introduce our approach below. Let us first consider taking expectations over any fixed policy $\pi$. Denote by $\delta$ the minimal probability over any action at any state for policies $\pi_1, \ldots, \pi_K$, i.e., $\delta = \min_{k \in [K]} \pi_k(a \mid s), \forall a \in \mathcal{A}, s \in \mathcal{S}$, the last term in (7) can be upper bounded by $\sum_{k=1}^{K} \mathbb{E}_{\pi}[\widetilde{D}_{\mathrm{KL}}(\pi_k^c \| \pi_k) - \widetilde{D}_{\mathrm{KL}}(\pi_k^c \| \pi_{k+1})] \leq H \log A + P_T \log \frac{1}{\delta}$, showing that we need to control the minimal value of $\delta$ to obtain a favorable dynamic regret bound. To this end, we slightly modify the update rule in (6) and add a uniform distribution $\pi^u(\cdot \mid s) = \frac{1}{A} \mathbf{1}, \forall s \in \mathcal{S}$. That is, the policy $\pi^u$ chooses each action with equal probability at any state. Thus, the update rule in (6) is modified as:

$$\pi'_{k,h}(\cdot \mid s) \propto \pi_{k-1,h}(\cdot \mid s) \exp(\eta \cdot Q_{k-1,h}^{\pi_{k-1}}(s, \cdot)), \pi_{k,h}(\cdot \mid s) = (1 - \gamma) \pi'_{k,h}(\cdot \mid s) + \gamma \pi^u(\cdot \mid s) \tag{8}$$

for any $s \in \mathcal{S}, h \in [H]$, where $\gamma \geq 0$ is the exploration parameter. This update is called the *fixed-share* mechanism in online learning literature [37]. While the fixed-share mechanism is standard to obtain dynamic regret in modern online learning [38], several important new challenges arise in online MDPs due to taking expectations over the policy sequence of *changing* policies $\pi_1^c, \ldots, \pi_K^c$. In particular, we prove that performing (8) ensures the following dynamic regret.

**Lemma 2.** *Set $\pi_1$ as uniform distribution on $\mathcal{A}$ for any state $s \in \mathcal{S}$. The update rule in* (8) *ensures*

$$\text{D-Regret}(K) \leq \frac{\eta K H^3}{2} + \frac{1}{\eta} \left( P_T \log \frac{A}{\gamma} + K H \log \frac{1}{1-\gamma} \right)$$

$$+ \frac{1}{\eta} \sum_{k=1}^{K} \sum_{h=1}^{H} \mathbb{E}_{\pi_k^c} \left[ \sum_{a \in \mathcal{A}} \left( \pi_{k-1,h}^c(a \mid s_h) \log \frac{1}{\pi'_{k,h}(a \mid s_h)} - \pi_{k,h}^c(a \mid s_h) \log \frac{1}{\pi'_{k+1,h}(a \mid s_h)} \right) \right] \tag{9}$$

The proof can be found in Appendix C.2. In the dynamic regret analysis in online learning, the last term in (9) is usually canceled out through telescoping since we do not need to take expectations [38].

However, this is *not* the case in online MDPs. Since the expectation is taken over the policy sequence of *changing* policies $\pi_1^c, \ldots, \pi_K^c$, this term cannot be canceled out, which requires a more refined analysis. To address this issue, we decompose one step of the expectation in (9) as follows.[1]

$$\left( \mathbb{E}_{\pi_{k-1}^c} \left[ \sum_{a \in \mathcal{A}} \pi_{k-1,h}^c \log \frac{1}{\pi_{k,h}'} \right] - \mathbb{E}_{\pi_k^c} \left[ \sum_{a \in \mathcal{A}} \pi_{k,h}^c \log \frac{1}{\pi_{k+1,h}'} \right] \right) + \mathbb{E}_{\pi_k^c - \pi_{k-1}^c} \left[ \sum_{a \in \mathcal{A}} \pi_{k-1,h}^c \log \frac{1}{\pi_{k,h}'} \right].$$

With this decomposition, the first term can be canceled out through telescoping, yet it remains to control the second term — the weighted difference between the state-action occupancy measures of policy $\pi_k^c$ and $\pi_{k-1}^c$ with weight $-\pi_{k-1,h}^c(a \mid s_h) \log \pi_{k,h}'(a \mid s_h)$ for state-action $(s_h, a)$. To control it, we need to (i) ensure the weight is upper bounded by some universal constant, and (ii) bound the unweighted difference between the state-action occupancy measures, which are new challenges that arose in online MDPs compared with standard online learning.

For the first challenge, note that the weight $-\pi_{k-1,h}^c(a \mid s_h) \log \pi_{k,h}'(a \mid s_h)$ can be large or even infinite since $\pi_{k,h}'$ is the policy before uniform exploration and $\pi_{k,h}'(a \mid s_h)$ can be arbitrarily small. Fortunately, $\pi_{k,h}'$ is obtained by one-step descent from $\pi_{k-1,h}$, which is the policy after uniform exploration and can be lower bounded. We provide the following *multiplicative stability lemma* for the one-step update, which shows $\pi_{k,h}'$ is not far from $\pi_{k-1,h}$ and thus is also lower bounded.

**Lemma 3** (Multiplicative Stability). *For any distributions $p \in \Delta(\mathcal{A})$ with $p(a) > 0$, for all $a \in A$, and any function $Q : \mathcal{S} \times \mathcal{A} \to [0, H]$, it holds for $p' \in \Delta(\mathcal{A})$ with $p'(a) \propto p(a) \cdot \exp(\eta \cdot Q(s, a))$ and $\eta \leq 1/H$ that $p'(a) \in [p(a)/4, 4p(a)]$, for all $a \in \mathcal{A}$.*

For the second challenge, we show the unweighted difference between the state-action occupancy measures can be bounded by the path length of policies. In particular, we have the following lemma.

**Lemma 4.** *For any policy sequence $\pi_1^c, \ldots, \pi_K^c$, it holds that*

$$\sum_{k=1}^{K} \left( \mathbb{E}_{\pi_k^c} - \mathbb{E}_{\pi_{k-1}^c} \right) \left[ \sum_{h=1}^{H} \mathbb{1}(s_h) \mid s_1 = s_{k,1} \right] \leq \sum_{k=1}^{K} \sum_{h=1}^{H} \sum_{i=1}^{h} \| \pi_{k,i}^c - \pi_{k-1,i}^c \|_{1,\infty} = H P_T.$$

**Remark 1.** We note that a similar argument is also used in Fei et al. [35, Appendix B.2.2]. However, they prove this lemma by imposing an additional smooth visitation measures assumption [35, Assumption 1], which is not required in our analysis.

The proofs for Lemmas 3 and 4 can be found in Appendices C.3 and C.4 respectively. Combining Lemmas 2, 3 and 4, we can prove the guarantee for update rule (8). The proof is in Appendix C.5.

**Theorem 1.** *Set $\pi_1$ as uniform distribution on $\mathcal{A}$ for any state $s \in \mathcal{S}$. The update rule in (8) ensures*

$$\text{D-Regret}(K) \leq \frac{\eta K H^3}{2} + \frac{1}{\eta} \left( H \log A + (1 + H) \log \frac{4A}{\gamma} P_T + K H \log \frac{1}{1 - \gamma} \right).$$

**Remark 2.** Considering the static regret where $\pi_1^c = \ldots = \pi_K^c = \pi^*$, we can recover the $\mathcal{O}(\sqrt{H^4 K \log A})$ static regret in Cai et al. [32] under the stationary scenario by setting $\gamma = 0$, that is, without uniform exploration. However, when $\gamma = 0$, the dynamic regret is not bounded as there lacks an upper bound for $-\log \gamma$, showing the necessity of the fixed-share mechanism.

### 4.2   Policy Evaluation Phase

Sections 4.1 focus on the simplified scenario where the transition is known. In this subsection, we further consider the unknown transition setting such that it is necessary to evaluate the policy $\pi_k$ based on the $(k-1)$ historical trajectories. To see how the model estimation error enters the dynamic regret, we decompose the dynamic regret in the following lemma.

**Lemma 5** (Fei et al. [35, Lemma 1]). *Define the model prediction error as $\iota_{k,h} = r_{k,h} + \mathbb{P}_h V_{k,h+1} - Q_{k,h}$, the dynamic regret $\text{D-Regret}(K) = \sum_{k=1}^{K} V_{k,1}^{\pi_k^c}(s_{k,1}) - \sum_{k=1}^{K} V_{k,1}^{\pi_k}(s_{k,1})$ can be written as*

$$\sum_{k,h} \mathbb{E}_{\pi_k^c} \left[ \langle Q_{k,h}(s_h, \cdot), \pi_{k,h}^c(\cdot \mid s_h) - \pi_{k,h}(\cdot \mid s_h) \rangle \right] + \mathcal{M}_{K,H} + \sum_{k,h} \left( \mathbb{E}_{\pi_k^c}[\iota_{k,h}(s_h, a_h)] - \iota_{k,h}(s_{k,h}, a_{k,h}) \right),$$

*where $\mathcal{M}_{K,H} = \sum_{k=1}^{K} \sum_{h=1}^{H} M_{k,h}$ is a martingale that satisfies $M_{k,h} \leq 4H, \forall k \in [k], h \in [H]$.*

---

[1]With a slight abuse of notations, we omit $(\cdot \mid s_h)$ for simplicity.

**Remark 3.** Lemma 5 is independent of the structure of MDPs. The first term in Lemma 5 is the dynamic regret over the estimated action-value function $Q_{k,h}$, which can be upper bounded by Theorem 5. The second term is a martingale, which can be bounded by Azuma-Hoeffding inequality. The third term is the model estimation error, which is the main focus of this section. Note the model prediction error $\iota_{k,h}(s_h, a_h)$ can be large for the state-action pairs that are less visited or even unseen. The general approach is incorporating the bonus function into the estimated $Q$-function such that $\iota_{k,h}(s_h, a_h) \leq 0$ for all $s \in \mathcal{S}, a \in \mathcal{A}$ (i.e., $\mathbb{E}_{\pi_k^c}[\iota_{k,h}(s_h, a_h)] \leq 0$) and we only need to control $-\iota_{k,h}(s_{k,h}, a_{k,h})$, which is the model estimation error at the visited state-action pair $(s_{k,h}, a_{k,h})$.

When applied to linear mixture MDPs, the key idea is learning the unknown parameter $\theta_h^*$ of the linear mixture MDP and using the learned parameter $\theta_{k,h}$ to build an optimistic estimator $Q_{k,h}(\cdot, \cdot)$ such that the model prediction error is non-positive, which is more or less standard. From the definition of linear mixture MDP, for the learned value function $V_{k,h}(\cdot)$, we have $[\mathbb{P}_h V_{k,h+1}](s, a) = \langle \sum_{s'} \phi(s' \mid s, a) V_{k,h+1}(s'), \theta_h^* \rangle = \langle \phi_{V_{k,h+1}}(s, a), \theta_h^* \rangle$. Inspired by recent advances in policy evaluation for linear mixture MDPs [31], we adopt the *weighted ridge regression* to estimate the parameter $\theta_h^*$, that is, we construct the estimator $\widehat{\theta}_{k,h}$ by solving the following weighted ridge regression problem:

$$\widehat{\theta}_{k,h} = \arg\min_{\theta \in \mathbb{R}^d} \sum_{j=1}^{k-1} \left[ \langle \phi_{V_{j,h+1}}(s_{j,h}, a_{j,h}), \theta \rangle - V_{j,h+1}(s_{j,h+1}) \right]^2 / \bar{\sigma}_{j,h}^2 + \lambda \|\theta\|_2^2.$$

Here, $\bar{\sigma}_{j,h}^2$ is the upper confidence bound of the variance $[\mathbb{V}_h V_{j,h+1}](s_{j,h}, a_{j,h})$, and we set it as $\bar{\sigma}_{k,h} = \sqrt{\max\{H^2/d, [\bar{\mathbb{V}}_{k,h} V_{k,h+1}](s_{k,h}, a_{k,h}) + E_{k,h}\}}$, where $[\bar{\mathbb{V}}_{k,h} V_{k,h+1}](s_{k,h}, a_{k,h})$ is a scalar-valued empirical estimate for the variance of the value function $V_{k,h+1}$ under the transition probability $\mathbb{P}_h(\cdot \mid s_k, a_k)$, and $E_{k,h}$ is the bonus term to guarantee that the true variance $[\mathbb{V}_{k,h} V_{k,h+1}](s_{k,h}, a_{k,h})$ is upper bounded by $[\bar{\mathbb{V}}_{k,h} V_{k,h+1}](s_{k,h}, a_{k,h}) + E_{k,h}$ with high probability. Then, the confidence set $\widehat{\mathcal{C}}_{k,h}$ is constructed as follows:

$$\widehat{\mathcal{C}}_{k,h} = \left\{ \theta \mid \|\widehat{\Sigma}_{k,h}^{1/2}(\theta - \widehat{\theta}_{k,h})\|_2 \leq \widehat{\beta}_k \right\}. \tag{10}$$

where $\widehat{\Sigma}_{k,h}$ is a covariance matrix based on the observed data, and $\widehat{\beta}_k$ is a radius of the confidence set. Given $\widehat{\mathcal{C}}_{k,h}$, we estimate the $Q$-function following the principle of "optimism in the face of uncertainty" [51] and set it as $Q_{k,h}(\cdot, \cdot) = [r_{k,h}(\cdot, \cdot) + \max_{\theta \in \widehat{\mathcal{C}}_{k,h}} \langle \theta, \phi_{V_{k,h+1}}(\cdot, \cdot) \rangle]_{[0,H-h+1]}$.

It remains to estimate the variance $[\mathbb{V}_h V_{k,h+1}](s_{k,h}, a_{k,h})$. By the definition of linear mixture MDPs, we have $[\mathbb{V}_h V_{k,h+1}](s_{k,h}, a_{k,h}) = \langle \phi_{V_{k,h+1}^2}(s_{k,h}, a_{k,h}), \theta_h^* \rangle - [\langle \phi_{V_{k,h+1}}(s_{k,h}, a_{k,h}), \theta_h^* \rangle]^2$. Therefore, we estimate $[\bar{\mathbb{V}}_{k,h} V_{k,h+1}](s_{k,h}, a_{k,h})$ by the expression below

$$\left[ \langle \phi_{V_{k,h+1}^2}(s_{k,h}, a_{k,h}), \widetilde{\theta}_{k,h} \rangle \right]_{[0,H^2]} - \left[ \langle \phi_{V_{k,h+1}}(s_{k,h}, a_{k,h}), \widehat{\theta}_{k,h} \rangle \right]_{[0,H]}^2, \tag{11}$$

where $\widetilde{\theta}_{k,h} = \arg\min_{\theta \in \mathbb{R}^d} \sum_{j=1}^{k-1} [\langle \phi_{V_{j,h+1}^2}(s_{j,h}, a_{j,h}), \theta \rangle - V_{j,h+1}^2(s_{j,h+1})]^2 + \lambda \|\theta\|_2^2$. The details are summarized in Lines 10-20 of Algorithm 1 and we provide the following guarantee.

**Theorem 2.** *Set the parameters as in Lemma 8, with probability at least $1 - \delta$, we have*

$$\sum_{k=1}^K \left( V_{k,1}(s_1^k) - V_{k,1}^{\pi^k}(s_1^k) \right) = \mathcal{M}_{K,H} - \sum_{k=1}^K \sum_{h=1}^H \iota_{k,h}(s_{k,h}, a_{k,h}) \leq \widetilde{\mathcal{O}}\left( \sqrt{dH^4K + d^2H^3K} \right).$$

The proof is given in Appendix C.6. Theorem 2 shows the model estimation error can be bounded. Combining Theorems 1, 2 and Lemma 5, we present the dynamic regret bound in the next section.

### 4.3 Regret Guarantee: Upper and Lower Bounds

In this section, we provide the regret bound for our algorithm and present a lower bound of the dynamic regret for any algorithm for adversarial linear mixture MDPs with the unknown transition.

**Theorem 3.** *Set $\eta = \min\{\sqrt{(P_T + \log A)/K}, 1\}/H$, $\gamma = 1/(KH)$ and $\widehat{\beta}_k$ as in Lemma 8, then with probability at least $1 - \delta$, it holds*

$$\text{D-Regret}(K) \leq \widetilde{\mathcal{O}}\left( \sqrt{dH^4K + d^2H^3K} + \sqrt{H^4(K + P_T)(1 + P_T)} \right), \tag{12}$$

*where $P_T = \sum_{k=1}^K \sum_{h=1}^H \|\pi_{k,h}^c - \pi_{k-1,h}^c\|_{1,\infty}$ is the path length of the compared policies.*

**Remark 4** (recovering static regret). Since static regret is a special case with $\pi_k^c = \pi^*, \forall k$, our result can recover the optimal $\widetilde{\mathcal{O}}(\sqrt{d^2 H^3 K})$ static regret when $H \leq d$, same as the result in He et al. [33].

**Remark 5** (improving linear mixture case). Our result improves upon the previously best-known $\widetilde{\mathcal{O}}(dH^{7/4}K^{3/4} + H^2 K^{2/3} P_T^{1/3})$ dynamic regret for adversarial linear mixture MDPs of Zhong et al. [36, Theorem 4.6] in terms of the dependence on $H$, $K$ and $P_T$.

**Remark 6** (improving tabular case). For the adversarial tabular MDPs, our result implies an $\widetilde{\mathcal{O}}(\sqrt{H^4 S^2 A K} + H^2 \sqrt{(K + P_T)(1 + P_T)})$ dynamic regret. This improves upon the best-known $\widetilde{\mathcal{O}}(\sqrt{H^4 S^2 A K} + H^2 K^{2/3} P_T^{1/3})$ result of Fei et al. [35, Theorem 1]. The details are in Appendix B.

We finally establish the lower bound of this problem. The proof can be found in Appendix C.8.

**Theorem 4.** *Suppose $B \geq 2, d \geq 4, H \geq 3, K \geq (d-1)^2 H/2$, for any algorithm and any constant $\Gamma \in [0, 2KH]$, there exists an episodic $B$-bounded adversarial linear mixture MDP and compared policies $\pi_1^c, \ldots, \pi_K^c$ such that $P_T \leq \Gamma$, and $\mathrm{D\text{-}Regret}(K) \geq \Omega(\sqrt{d^2 H^3 K} + \sqrt{HK(H + \Gamma)})$.*

When $H \leq d$, the upper bound is $\widetilde{\mathcal{O}}(\sqrt{d^2 H^3 K} + \sqrt{H^4(K + P_T)(1 + P_T)})$. Combining it with Theorem 4, we discuss the optimality of our result. We consider the following three regimes.

- Small $P_T$: when $0 \leq P_T \leq d^2/H$, the upper bound (12) can be simplified as $\widetilde{\mathcal{O}}(\sqrt{d^2 H^3 K})$, and the lower bound is $\widetilde{\mathcal{O}}(\sqrt{d^2 H^3 K})$, hence our result is optimal in terms of $d$, $H$ and $K$.

- Moderate $P_T$: when $d^2/H \leq P_T \leq K$, the upper bound (12) can be simplified as $\widetilde{\mathcal{O}}(\sqrt{d^2 H^3 K} + \sqrt{H^4 K(1 + P_T)})$, and it is minimax optimal in $d$, $K$ and $P_T$ but looses a factor of $H^{\frac{3}{2}}$.

- Large $P_T$: when $P_T \geq K$, any algorithm suffers at most $\mathcal{O}(HK)$ dynamic regret, while the lower bound is $\Omega(K\sqrt{H})$. So our result is minimax optimal in $K$ but looses a factor of $\sqrt{H}$.

## 5 Towards Optimal Dynamic Regret with Unknown $P_T$

This section further considers the case when the non-stationarity measure $P_T$ is unknown. By Theorem 1, we need to tune the step size $\eta$ optimally to balance the number of episodes $K$ and $P_T$ to achieve a favorable dynamic regret. To address the difficulty of not knowing $P_T$ ahead of time, we develop an online ensemble method to handle this uncertainty, in which a two-layer meta-base structure is maintained. While the methodology can be standard in recent non-stationary online learning [52, 39, 40, 41], new challenges arise in online MDPs. We introduce the details below.

By the performance difference lemma in Cai et al. [32, Lemma 3.2] (as restate in Lemma 13), we can rewrite the dynamic regret as

$$\sum_{k=1}^{K} \left[ V_{k,1}^{\pi_k^c}\left(s_1^k\right) - V_{k,1}^{\pi_k}\left(s_1^k\right) \right] = \sum_{k=1}^{K} \mathbb{E}_{\pi_k^c} \left[ \sum_{h=1}^{H} \left\langle Q_{k,h}^{\pi_k}(s_h, \cdot), \pi_{k,h}^c(\cdot \mid s_h) - \pi_{k,h}(\cdot \mid s_h) \right\rangle \right],$$

where the expectation is taken over the randomness of the state trajectory sampled according to $\pi_k^c$. The dynamic regret of online MDPs can be written as the weighted average of some "multi-armed bandits" problems over all states, where the weight for each state is the *unknown* and *changing* probability of being visited by $\pi_1^c, \ldots, \pi_K^c$. As the optimal step size depends on the unknown non-stationarity measure $P_T$ as shown in Section 4, a natural idea is to the two-layer structure to learn the optimal step size as in recent online convex optimization literature [52, 39, 40, 41].

The general idea is constructing a step size pool $\mathcal{H} = \{\eta_1, \ldots, \eta_N\}$ to discretize the value range of the optimal step size; and then maintaining multiple base-learners $\mathcal{B}_1, \ldots, \mathcal{B}_N$, each of which works with a specific step size $\eta_i$. Finally, a meta-algorithm is used to track the best base-learner and yield the final policy. Then, the dynamic regret can be decomposed as follows (omit $(\cdot \mid s_h)$ for simplicity):

$$\sum_{k=1}^{K} \mathbb{E}_{\pi_k^c} \left[ \sum_{h=1}^{H} \left\langle Q_{k,h}^{\pi_k}, \pi_{k,h}^c - \pi_{k,h}^i \right\rangle \right] + \sum_{k=1}^{K} \mathbb{E}_{\pi_k^c} \left[ \sum_{h=1}^{H} \left\langle Q_{k,h}^{\pi_k}, \pi_{k,h}^i - \pi_{k,h} \right\rangle \right].$$

Since the above decomposition holds for any index $i \in [N]$, we can always choose the base-learner with optimal step size to analyze and the first term is easy to control. The challenge is to control the

second term, which is the regret of the meta-algorithm. Different from the standard "Prediction with Expert Advice" problem, it involves an additional expectation over the randomness of states sampled according to $\pi_k^c$. This poses a grand challenge compared to conventional online convex optimization where the expectation is not required. Although we can bound this term by $P_T$ again, optimal tuning of the meta-algorithm is hindered as $P_T$ is unknown. Consequently, we opt to upper bound it by the worst-case dynamic regret [53], that is, benchmarking the performance with the best choice of each round, which in turn introduces the dependence on the switching number of the best base-learner.

We introduce our approach as follows. We maintain multiple base-learners, each of which works with a specific step size $\eta_i$. All base-learners update their policies according to the same action-value function $Q_{k-1,h}(s_h, \cdot)$ of the combined policy $\pi_{k-1}$, that is, the base-learner $\mathcal{B}_i$ updates policy by

$$\pi_{k,h}^{i,\prime}(\cdot \mid s) \propto \pi_{k-1,h}^i(\cdot \mid s)\exp(\eta_i Q_{k-1,h}(s, \cdot)), \pi_{k,h}^i(\cdot \mid s) = (1-\gamma)\pi_{k,h}^{i,\prime}(\cdot \mid s) + \gamma\pi^u(\cdot \mid s), \quad (13)$$

Then, the meta-algorithm chooses the base-learner by measuring the quality of each base-learner. In our approach, we choose the best base-learner at the last episode, that is,

$$\pi_{k,h}(\cdot \mid s) = \pi_{k,h}^{i_{k-1,h}^*}(\cdot \mid s) \text{ with } i_{k-1,h}^*(s) = \arg\max_{i\in[N]}\langle Q_{k-1,h}(s, \cdot), \pi_{k-1,h}^i(\cdot \mid s)\rangle. \quad (14)$$

The details are summarized in Algorithm 2 of Appendix A and the guarantee is as follows.

**Theorem 5.** *Set $\gamma = 1/(KH)$, step size pool $\mathcal{H} = \{\eta_i = (2^i/H)\sqrt{(\log A)/K} \mid i \in [N]\}$ with $N = \lfloor\frac{1}{2}\log(\frac{K}{\log A})\rfloor$. Algorithm 2 ensures*

$$\text{D-Regret}(K) \leq \widetilde{\mathcal{O}}\Big(\sqrt{dH^4 K + d^2 H^3 K} + \sqrt{H^4(K + P_T)(1 + P_T) + H^2 S_T^2}\Big),$$

*where $P_T = \sum_{k=1}^K \sum_{h=1}^H \|\pi_{k,h}^c - \pi_{k-1,h}^c\|_{1,\infty}$ is the path length of the compared policies, $S_T = \sum_{k=1}^K \sum_{h=1}^H \mathbb{E}_{\pi_k^c} \mathbb{1}[i_{k,h}^*(s_h) \neq i_{k-1,h}^*(s_h)]$ is the expected switching number of best base-learner.*

Combining it with Theorem 3, we discuss the optimality of our result. We consider two regimes.

- Small $S_T$: when $S_T \leq \max\{d\sqrt{HK}, H\sqrt{(K + P_T)(1 + P_T)}\}$, the term $S_T$ can be subsumed by other terms. In this case, the upper bound in Theorem 5 is entirely the *same* as that in Theorem 3. This implies we maintain the same guarantees without $P_T$ as algorithmic input.
- Large $S_T$: when $S_T > \max\{d\sqrt{HK}, H\sqrt{(K + P_T)(1 + P_T)}\}$, our result looses a factor of $HS_T$ compared with the result in Theorem 3 for the known $P_T$ setting.

By the above discussion, our result can be optimal in terms of $K$ and $P_T$ under certain regimes when $P_T$ is unknown. In comparison, the regret bounds achieved via the restart mechanism [35, 36] remain sub-optimal across all regimes even $P_T$ is known. Note that we introduce the notation $S_T$ in the regret analysis, which also reflects the degree of environmental non-stationarity to some extent. Consider the following two examples: (i) in the stationary environment, $S_T$ could be relatively small as the best base-learner would seldom change, and (ii) in the piecewise-stationary environment, $S_T$ would align with the frequency of environmental changes. Indeed, given that $S_T$ is a data-dependent quantity, its inclusion in the regret analysis is not ideal. Deriving bounds that rely exclusively on problem-dependent quantities, like $P_T$, remains a significant open challenge.

## 6 Conclusion and Future Work

In this work, we study the dynamic regret of adversarial linear mixture MDPs with the unknown transition. For the case when $P_T$ is known, we propose a novel policy optimization algorithm that incorporates a *fixed-share* mechanism without the need for restarts. We show it enjoys a dynamic regret of $\widetilde{\mathcal{O}}\big(\sqrt{d^2 H^3 K} + \sqrt{H^4(K + P_T)(1 + P_T)}\big)$, strictly improving the previously best-known result of Zhong et al. [36] for the same setting and Fei et al. [35] when specialized to tabular MDPs. We also establish an $\Omega\big(\sqrt{d^2 H^3 K} + \sqrt{HK(H + P_T)}\big)$ lower bound, indicating that our algorithm is optimal regarding $d$, $K$ and $P_T$ and can be optimal in terms of $H$ under certain regimes. Moreover, we explore the more complex scenario where $P_T$ is unknown. We show this setting presents unique challenges that distinguish online MDPs from conventional online convex optimization. We introduce a novel two-layer algorithm and show its dynamic regret guarantee is attractive under certain regimes.

There are several important future works to investigate. First, how to remove the dependence on the switching number $S_T$ is an important open question. Moreover, we focus on the full-information feedback in this work, it remains an open problem to extend the results to the bandit feedback.

## Acknowledgements

This research was supported by National Key R&D Program of China (2022ZD0114800) and NSFC (62206125, 61921006). Peng Zhao was supported in part by the Xiaomi Foundation.

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

# A    Algorithm with Unknown $P_T$

In this part, we present the POWERS-FixShare-OnE algorithm in Algorithm 2 for the case when $P_T$ is unknown. The algorithm is based on the POWERS-FixShare algorithm and we further employ an online ensemble structure to eliminate the algorithmic dependence on $P_T$. In Line 5, each base-learner updates her policy with a specific step size $\eta_i$ and the meta-learner selects the best policy among the base-learners in Line 6. In the policy evaluation phase (Line 11 - 21), we use the same estimator as in Algorithm 1 to estimate the parameter $\theta_h^*$ and construct the confidence set.

---

**Algorithm 2** POWERS-FixShare-OnE

---

**Input:** step size pool $\mathcal{H}$, exploration parameter $\gamma$ and regularization parameter $\lambda$.

1: Initialize $\{\pi_{0,h}^i(\cdot \mid s)\}_{h=1}^H, \forall i \in [N], s \in \mathcal{S}$ as uniform distribution on $\mathcal{A}$, $\{p_{0,h}(s)\}_{h=1}^H, \forall s \in \mathcal{S}$
   as uniform distribution on $N$ and set $\{Q_{0,h}(\cdot, \cdot)\}_{h=1}^H$ as zero function.

2: **for** $k = 1, 2, \cdots, K$ **do**

3:    Receive the initial state $s_{k,1}$.

4:    **for** $h = 1, 2, \cdots, H$ **do**

5:       For all $h \in [H], s \in \mathcal{S}$, each base-learner $\mathcal{B}_i, \forall i \in [N]$ updates policy by

$$\pi_{k,h}^{i,\prime}(\cdot \mid s) \propto \pi_{k-1,h}^i(\cdot \mid s) \exp\left(\eta_i Q_{k-1,h}(s, \cdot)\right), \pi_{k,h}^i(\cdot \mid s) = (1 - \gamma)\pi_{k,h}^{i,\prime}(\cdot \mid s) + \gamma\pi^u(\cdot \mid s).$$

6:       Set $\pi_{k,h}(\cdot \mid s) = \pi_{k,h}^{i_{k-1,h}^*}(\cdot \mid s)$ with $i_{k-1,h}^*(s) = \arg\max_{i \in [N]}\langle Q_{k-1,h}(s, \cdot), \pi_{k-1,h}^i(\cdot \mid s)\rangle$.

7:       Take the action following $a_{k,h} \sim \pi_{k,h}(\cdot \mid s_{k,h})$ and transit to the next state $s_{k,h+1}$.

8:       Obtain reward $r_{k,h}(s_{k,h}, a_{k,h})$ and observe the reward function $r_{k,h}(\cdot, \cdot)$.

9:    **end for**

10:   Initialize $V_{k,H+1}(\cdot)$ as a zero function.

11:   **for** $h = H, H-1, \cdots, 1$ **do**

12:      Set $Q_{k,h}(\cdot, \cdot) \leftarrow \left[r_{k,h}(\cdot, \cdot) + \langle\widehat{\theta}_{k,h}, \phi_{V_{k,h+1}}(\cdot, \cdot)\rangle + \widehat{\beta}_k\|\widehat{\Sigma}_{k,h}^{-1/2}\phi_{V_{k,h+1}}(\cdot, \cdot)\|_2\right]_{[0,H-h+1]}$.

13:      Set $V_{k,h}(\cdot) \leftarrow \mathbb{E}_{a \sim \pi_{k,h}(\cdot \mid \cdot)}[Q_{k,h}(\cdot, \cdot)]$.

14:      Set the estimated variance $\left[\bar{\mathbb{V}}_{k,h}V_{k,h+1}\right](s_{k,h}, a_{k,h})$ as in (11), bonus $E_{k,h}$ as in (21).

15:      $\bar{\sigma}_{k,h} \leftarrow \sqrt{\max\left\{H^2/d, \left[\bar{\mathbb{V}}_{k,h}V_{k,h+1}\right](s_{k,h}, a_{k,h}) + E_{k,h}\right\}}$.

16:      $\widehat{\Sigma}_{k+1,h} \leftarrow \widehat{\Sigma}_{k,h} + \bar{\sigma}_{k,h}^{-2}\phi_{V_{k,h+1}}(s_{k,h}, a_{k,h})\phi_{V_{k,h+1}}(s_{k,h}, a_{k,h})^\top$.

17:      $\widehat{b}_{k+1,h} \leftarrow \widehat{b}_{k,h} + \bar{\sigma}_{k,h}^{-2}\phi_{V_{k,h+1}}(s_{k,h}, a_{k,h})V_{k,h+1}(s_{h+1}^k)$.

18:      $\widetilde{\Sigma}_{k+1,h} \leftarrow \widetilde{\Sigma}_{k,h} + \phi_{V_{k,h+1}^2}(s_{k,h}, a_{k,h})\phi_{V_{k,h+1}^2}(s_{k,h}, a_{k,h})^\top$.

19:      $\widetilde{b}_{k+1,h} \leftarrow \widetilde{b}_{k,h} + \phi_{V_{k,h+1}^2}(s_{k,h}, a_{k,h})V_{k,h+1}^2(s_{h+1}^k)$.

20:      $\widehat{\theta}_{k+1,h} \leftarrow \widehat{\Sigma}_{k+1,h}^{-1}\widehat{b}_{k+1,h}, \widetilde{\theta}_{k+1,h} \leftarrow \widetilde{\Sigma}_{k+1,h}^{-1}\widetilde{b}_{k+1,h}$.

21:   **end for**

22: **end for**

---

# B    Recovering Tabular Case

In this part, we show the result of our algorithm when specialized to the tabular case. Note in the linear case, we adopt the *weighted ridge regression* to estimate the parameter $\theta_h^*$, that is, we construct the estimator $\widehat{\theta}_{k,h}$ by solving the following weighted ridge regression problem:

$$\widehat{\theta}_{k,h} = \underset{\theta \in \mathbb{R}^d}{\arg\min} \sum_{j=1}^{k-1}\left[\langle\phi_{V_{j,h+1}}(s_{j,h}, a_{j,h}), \theta\rangle - V_{j,h+1}(s_{j,h+1})\right]^2 / \bar{\sigma}_{j,h}^2 + \lambda\|\theta\|_2^2.$$

In the tabular case, we simply set $\bar{\sigma}_{j,h} = 1$, and compute $\phi_{V_{k,h+1}}$ by taking the sample mean of $\{V_{k,h+1}(s_{j,h+1})\}_{j \in [k-1]}$. That is, we set it as

$$\phi_{V_{j,h+1}}(s_{j,h}, a_{j,h}) = \sum_{s' \in \mathcal{S}}\frac{N_k(s_{j,h}, a_{j,h}, s')}{N_k(s_{j,h}, a_{j,h}) + \lambda} \cdot V_{j,h+1}(s'),$$

of each $(s, a)$, where $N_k$ counts the number of times each tuple $(s, a, s)$ or $(s, a)$ has been visited up to episode $k$. Then, we construct the estimator $\widehat{\theta}_{k,h}$ by solving the linear regression problem:

$$\widehat{\theta}_{k,h} = \underset{\theta \in \mathbb{R}^d}{\arg\min} \sum_{j=1}^{k-1} \left[ \langle \phi_{V_{j,h+1}}(s_{j,h}, a_{j,h}), \theta \rangle - V_{j,h+1}(s_{j,h+1}) \right]^2 + \lambda \|\theta\|_2^2.$$

Then, the confidence set $\widehat{\mathcal{C}}_{k,h}$ is constructed as follows:

$$\widehat{\mathcal{C}}_{k,h} = \left\{ \theta \mid \|\widehat{\Sigma}_{k,h}^{1/2}(\theta - \widehat{\theta}_{k,h})\|_2 \leq \beta \right\}.$$

The remaining part of the algorithm is the same as the linear case. The following theorem shows the result of our algorithm in the tabular case.

**Theorem 6.** *Set* $\gamma = 1/(KH)$, *step size pool* $\mathcal{H} = \{\eta_i = (2^i/H)\sqrt{(\log A)/K} \mid i \in [N]\}$ *with* $N = \lfloor \frac{1}{2} \log(\frac{K}{\log A}) \rfloor$ *and* $\beta = H\sqrt{S \log(dKH/\delta)}$, *with probability at least* $1 - \delta$, *it holds*

$$\text{D-Regret}(K) \leq \widetilde{\mathcal{O}}\left( \sqrt{H^4 S^2 A K} + \sqrt{H^4 (K + P_T)(1 + P_T)} \right),$$

*where* $P_T = \sum_{k=1}^{K} \sum_{h=1}^{H} \|\pi_{k,h}^c - \pi_{k-1,h}^c\|_{1,\infty}$ *is the path length of the compared policy sequence.*

*Proof.* By Lemma 5, we decompose the dynamic regret as follows.

$$\begin{aligned}
\text{D-Regret}(K) \leq &\sum_{k=1}^{K} \sum_{h=1}^{H} \mathbb{E}_{\pi_k^c} \left[ \langle Q_{k,h}(s_h, \cdot), \pi_{k,h}^c(\cdot \mid s_h) - \pi_{k,h}(\cdot \mid s_h) \rangle \right] + \mathcal{M}_{K,H} \\
&+ \sum_{k=1}^{K} \sum_{h=1}^{H} \left( \mathbb{E}_{\pi_k^c}[\iota_{k,h}(s_h, a_h)] - \iota_{k,h}(s_{k,h}, a_{k,h}) \right),
\end{aligned}$$

Note that the policy evaluation phase is the same as the one in Fei et al. [35, Algorithm 3], by the estimator bound in Fei et al. [35, Lemmas 7 and 9], we have

$$\sum_{k=1}^{K} \sum_{h=1}^{H} \mathbb{E}_{\pi_k^c}[\iota_{k,h}(s_h, a_h)] \leq 0, \sum_{k=1}^{K} \sum_{h=1}^{H} -\iota_{k,h}(s_{k,h}, a_{k,h}) \leq \mathcal{O}\left( \sqrt{H^4 S^2 A T \log^2(SAKH/\delta)} \right). \tag{15}$$

The remaining proof is the same as the proof of Theorem 3 in Appendix C.7. ■

# C Proofs

In this section, we provide the proof of the results in this paper.

## C.1 Proof of Lemma 1

*Proof.* By the definition of dynamic regret, we have

$$\begin{aligned}
&\text{D-Regret}(K) \\
&= \sum_{k=1}^{K} V_{k,1}^{\pi_k^c}(s_{k,1}) - \sum_{k=1}^{K} V_{k,1}^{\pi_k}(s_{k,1}) \\
&= \sum_{k=1}^{K} \sum_{h=1}^{H} \mathbb{E}_{\pi_k^c} \left[ \langle Q_{k,h}^{\pi_k}(s_h, \cdot), \pi_{k,h}^c(\cdot \mid s_h) - \pi_{k,h}(\cdot \mid s_h) \rangle \mid s_1 = s_{k,1} \right] \\
&\leq \frac{\eta K H^3}{2} + \sum_{k=1}^{K} \sum_{h=1}^{H} \mathbb{E}_{\pi_k^c} \left[ D_{\text{KL}}\left( \pi_{k,h}^c(\cdot \mid s_h) \| \pi_{k,h}(\cdot \mid s_h) \right) - D_{\text{KL}}\left( \pi_{k,h}^c(\cdot \mid s_h) \| \pi_{k+1,h}(\cdot \mid s_h) \right) \right],
\end{aligned}$$

where the equality holds by the performance difference lemma in Lemma 13 and the inequality holds by the one-step descent guarantee in Lemma 14. This finishes the proof. ■

## C.2 Proof of Lemma 2

*Proof.* Note the update rule is

$$\pi'_{k,h}(\cdot \mid s) \propto \pi_{k-1,h}(\cdot \mid s) \exp(\eta \cdot Q^{\pi_{k-1}}_{k-1,h}(s, \cdot)), \pi_{k,h}(\cdot \mid s) = (1 - \gamma)\pi'_{k,h}(\cdot \mid s) + \gamma\pi^u(\cdot \mid s)$$

By Lemma 14, for any $s_h \in \mathcal{S}, h \in [H], k \in [K]$, we have

$$\langle Q^{\pi_k}_{k,h}(s_h, \cdot), \pi^c_{k,h}(\cdot \mid s_h) - \pi_{k,h}(\cdot \mid s_h) \rangle$$

$$\leq \frac{\eta H^2}{2} + \frac{1}{\eta} \left( D_{\mathrm{KL}}\left( \pi^c_{k,h}(\cdot \mid s_h) \| \pi_{k,h}(\cdot \mid s_h) \right) - D_{\mathrm{KL}}\left( \pi^c_{k,h}(\cdot \mid s_h) \| \pi'_{k+1,h}(\cdot \mid s_h) \right) \right)$$

$$= \frac{\eta H^2}{2} + \frac{1}{\eta} \sum_{a \in \mathcal{A}} \left( \pi^c_{k,h}(a \mid s_h) \log \frac{1}{\pi_{k,h}(a \mid s_h)} - \pi^c_{k,h}(a \mid s_h) \log \frac{1}{\pi'_{k+1,h}(a \mid s_h)} \right), \qquad (16)$$

where the equality holds by the definition of KL divergence. We decompose the last term as follows:

$$\sum_{a \in \mathcal{A}} \left( \pi^c_{k,h}(a \mid s_h) \log \frac{1}{\pi_{k,h}(a \mid s_h)} - \pi^c_{k,h}(a \mid s_h) \log \frac{1}{\pi'_{k+1,h}(a \mid s_h)} \right)$$

$$= \sum_{a \in \mathcal{A}} \left( \pi^c_{k,h}(a \mid s_h) \log \frac{1}{\pi_{k,h}(a \mid s_h)} - \pi^c_{k-1,h}(a \mid s_h) \log \frac{1}{\pi'_{k,h}(a \mid s_h)} \right) \qquad (17)$$

$$+ \sum_{a \in \mathcal{A}} \left( \pi^c_{k-1,h}(a \mid s_h) \log \frac{1}{\pi'_{k,h}(a \mid s_h)} - \pi^c_{k,h}(a \mid s_h) \log \frac{1}{\pi'_{k+1,h}(a \mid s_h)} \right).$$

For the first term, we have

$$\sum_{a \in \mathcal{A}} \left( \pi^c_{k,h}(a \mid s_h) \log \frac{1}{\pi_{k,h}(a \mid s_h)} - \pi^c_{k-1,h}(a \mid s_h) \log \frac{1}{\pi'_{k,h}(a \mid s_h)} \right)$$

$$= \sum_{a \in \mathcal{A}} \left( (\pi^c_{k,h}(a \mid s_h) - \pi^c_{k-1,h}(a \mid s_h)) \log \frac{1}{\pi_{k,h}(a \mid s_h)} + \pi^c_{k-1,h}(a \mid s_h) \log \frac{\pi'_{k,h}(a \mid s_h)}{\pi_{k,h}(a \mid s_h)} \right).$$

By the update rule, we have $1 \leq 1/\pi_{k,h}(a \mid s_h) \leq A/\gamma$ and $\pi'_{k,h}(a \mid s_h)/\pi_{k,h}(a \mid s_h) \leq 1/(1 - \gamma)$. Therefore, we have

$$\sum_{a \in \mathcal{A}} \left( (\pi^c_{k,h}(a \mid s_h) - \pi^c_{k-1,h}(a \mid s_h)) \log \frac{1}{\pi_{k,h}(a \mid s_h)} + \pi^c_{k-1,h}(a \mid s_h) \log \frac{\pi'_{k,h}(a \mid s_h)}{\pi_{k,h}(a \mid s_h)} \right)$$

$$\leq \|\pi^c_{k,h}(\cdot \mid s_h) - \pi^c_{k-1,h}(\cdot \mid s_h)\|_1 \cdot \log \frac{A}{\gamma} + \log \frac{1}{1 - \gamma}. \qquad (18)$$

Then, the dynamic regret is bounded as follows:

$$\text{D-Regret}(K)$$

$$= \sum_{k=1}^{K} V^{\pi^c_k}_{k,1}(s_{k,1}) - \sum_{k=1}^{K} V^{\pi_k}_{k,1}(s_{k,1})$$

$$= \sum_{k=1}^{K} \sum_{h=1}^{H} \mathbb{E}_{\pi^c_k} \left[ \langle Q^{\pi_k}_{k,h}(s_h, \cdot), \pi^c_{k,h}(\cdot \mid s_h) - \pi_{k,h}(\cdot \mid s_h) \rangle \mid s_1 = s_{k,1} \right]$$

$$\leq \frac{\eta K H^3}{2} + \frac{1}{\eta} \sum_{k=1}^{K} \sum_{h=1}^{H} \sum_{a \in \mathcal{A}} \mathbb{E}_{\pi^c_k} \left[ \pi^c_{k,h}(a \mid s_h) \log \frac{1}{\pi_{k,h}(a \mid s_h)} - \pi^c_{k,h}(a \mid s_h) \log \frac{1}{\pi'_{k+1,h}(a \mid s_h)} \right]$$

$$\leq \frac{\eta K H^3}{2} + \frac{1}{\eta} \sum_{k=1}^{K} \sum_{h=1}^{H} \mathbb{E}_{\pi^c_k} \left[ \|\pi^c_{k,h}(\cdot \mid s_h) - \pi^c_{k-1,h}(\cdot \mid s_h)\|_1 \cdot \log \frac{A}{\gamma} + \log \frac{1}{1 - \gamma} \right]$$

$$+ \sum_{k=1}^{K} \sum_{h=1}^{H} \mathbb{E}_{\pi^c_k} \left[ \sum_{a \in \mathcal{A}} \left( \pi^c_{k-1,h}(a \mid s_h) \log \frac{1}{\pi'_{k,h}(a \mid s_h)} - \pi^c_{k,h}(a \mid s_h) \log \frac{1}{\pi'_{k+1,h}(a \mid s_h)} \right) \right]$$

$$\leq \frac{\eta K H^3}{2} + \frac{1}{\eta}\left(P_T \log\frac{A}{\gamma} + KH\log\frac{1}{1-\gamma}\right)$$
$$+ \sum_{k=1}^{K}\sum_{h=1}^{H}\mathbb{E}_{\pi_k^c}\left[\sum_{a\in\mathcal{A}}\left(\pi_{k-1,h}^c(a\mid s_h)\log\frac{1}{\pi_{k,h}'(a\mid s_h)} - \pi_{k,h}^c(a\mid s_h)\log\frac{1}{\pi_{k+1,h}'(a\mid s_h)}\right)\right],$$

where the first inequality holds by (16), the second inequality is due to (17) and (18), and the last inequality holds by the definition of $P_T = \sum_{k=1}^{K}\sum_{h=1}^{H}\|\pi_{k,h}^c - \pi_{k-1,h}^c\|_{1,\infty}$. It ends the proof. $\blacksquare$

### C.3  Proof of Lemma 3

*Proof.* Lemma 3 is a simplified version of Chen et al. [54, Lemma 17] and we prove it in our notations below for self-containedness. It is easy to verify that the update rule $p'(a) \propto p(a)\cdot\exp(\eta\cdot Q(s,a))$ is equivalent to the update

$$p' = \arg\max_{p'\in\Delta(\mathcal{A})} \eta\langle p', Q(s,\cdot)\rangle + D_{\mathrm{KL}}(p'\|p).$$

Thus, by the KKT condition, we have for some $\lambda$ and $\mu(a)\geq 0$, such that

$$Q(s,a) - \frac{1}{\eta}\log\frac{p'(a)}{p(a)} + \lambda + \mu(a) = 0, \text{ and } \mu(a)p'(a) = 0, \forall a\in\mathcal{A}.$$

The above equations give the closed-form solution $p'(a) = p(a)\exp(\eta(Q(s,a)+\lambda+\mu(a)))$. First, we prove that for all $a\in\mathcal{A}$ we have $\mu(a) = 0$. Indeed, when $\mu(a)\neq 0$, by $\mu(a)p'(a) = 0$, we have $p'(a) = p(a)\exp(\eta(Q(s,a)+\lambda+\mu(a))) = 0$, which contradicts with $p(a) > 0$.

Then, we now separately discuss two cases.

**Case 1:** $\min_{a\in\mathcal{A}}Q(s,a)\neq\max_{a\in\mathcal{A}}Q(s,a)$.  In this case, we first show that $\min_{a\in\mathcal{A}}-Q(s,a)\leq\lambda\leq\max_{a\in\mathcal{A}}-Q(s,a)$. We prove it by contradiction: If $\lambda\geq\max_{a\in\mathcal{A}}-Q(s,a)$, then we have

$$\sum_{a\in\mathcal{A}}p'(a) = \sum_{a\in\mathcal{A}}p(a)\exp(\eta(Q(s,a)+\lambda+\mu(a))) \geq \sum_{a\in\mathcal{A}}p(a) \geq 1.$$

contradicting with $p'\in\Delta(\mathcal{A})$. A similar argument holds for the case $\lambda\leq\min_{a\in\mathcal{A}}(-Q(s,i))$. Thus, we have $\lambda\in[\min_{a\in\mathcal{A}}(-Q(s,a)),\max_{a\in\mathcal{A}}(-Q(s,a))]$. Then, we have $|Q(s,a)+\lambda+\mu(a)|\leq\max_{a\in\mathcal{A}}Q(s,a)-\min_{a\in\mathcal{A}}Q(s,a)\leq H$. By the condition on $\eta H\leq 1$, we have $p'(a)\in[\exp(-1)p(a),\exp(1)p(a)]\in[1/(4p(a)),4p(a)]$.

**Case 2:** $\min_{a\in\mathcal{A}}Q(s,a)=\max_{a\in\mathcal{A}}Q(s,a)$.  In this case, it is clear that $\lambda=-Q(s,a)$ must hold for all $a\in\mathcal{A}$ to make $p$ and $p'$ both discussions. Thus $p'(a)=p(a)$ for all $a\in\mathcal{A}$.

Combining the above two cases finishes the proof. $\blacksquare$

### C.4  Proof of Lemma 4

To prove Lemma 4, we first introduce the following two lemmas. Denote by $\mathbb{P}_h^{\pi_h}(s'\mid s) = \sum_{a\in\mathcal{A}}\mathbb{P}_h(s'\mid s,a)\pi_h(a\mid s)$ the transition kernel of the MDP in step $h$ under policy $\pi_h$ and recall that $\|\pi-\pi'\|_{1,\infty} = \max_{s\in\mathcal{S}}\|\pi(\cdot\mid s)-\pi'(\cdot\mid s)\|$. The first lemma shows the difference between the state distribution of two policies can be bounded by the path length of the policies.

**Lemma 6** (Zhao et al. [34, Lemma 7]). *For any state distribution $d$, policy pair $\pi$ and $\pi'$, and transition kernel $\mathbb{P}$, we have*

$$\left\|d\mathbb{P}_h^{\pi_h}(\cdot,s) - d\mathbb{P}_h^{\pi_h'}(\cdot,s)\right\|_1 \leq \|\pi_h-\pi_h'\|_{1,\infty}, \forall h\in[H].$$

*Proof.* Consider the case when $d$ is a delta function on $s$. The difference in the next distributions is

$$\left\|\mathbb{P}_h^{\pi_h}(\cdot,s) - \mathbb{P}_h^{\pi_h}(\cdot,s)\right\|_1 = \sum_{s'\in\mathcal{S}}\sum_{a\in\mathcal{A}}|\mathbb{P}(s'\mid s,a)(\pi_h(a\mid s)-\pi_h(a\mid s))|$$
$$\leq \sum_{s'\in\mathcal{S}}\sum_{a\in\mathcal{A}}\mathbb{P}(s'\mid s,a)\|\pi_h(a\mid s)-\pi_h'(a\mid s)\|_1$$

$$\le \sum_{a \in \mathcal{A}} |\pi(a \mid s) - \pi'(a \mid s)| \le \|\pi_h - \pi'_h\|_{1,\infty}.$$

Linearity of expectation leads to the result for arbitrary distributions. This finishes the proof. ∎

The second lemma shows the difference between the state distribution of the policy starting from the different initial distributions can be bounded by the difference between the initial distributions.

**Lemma 7** (Zhao et al. [34, Lemma 8]). *For any two initial distributions $d$ and $d'$, transition kernel $\mathbb{P}$ and policy $\pi$, we have*

$$\left\| d\mathbb{P}_h^{\pi_h} - d'\mathbb{P}_h^{\pi_h} \right\|_1 \le \|d - d'\|_1, \forall h \in [H].$$

*Proof.* Note the relationship that $d(s') = \sum_{s \in \mathcal{S}} d(s)\mathbb{P}_{s,s'}^{\pi_h}$, we have

$$
\begin{aligned}
\left\| d\mathbb{P}_h^{\pi_h} - d'\mathbb{P}_h^{\pi_h} \right\|_1 &= \sum_{s'} \left| \sum_s d(s)\mathbb{P}_h^{\pi_h}(s' \mid s) - d'(s)\mathbb{P}_h^{\pi_h}(s' \mid s) \right| \\
&\le \sum_{s'} \sum_s |d(s)\mathbb{P}_h^{\pi_h}(s' \mid s) - d'(s)\mathbb{P}_h^{\pi_h}(s' \mid s)| \\
&= \sum_{s'} \sum_s |d(s) - d'(s)| \, \mathbb{P}_h^{\pi_h}(s' \mid s) \\
&= \sum_s |d(s) - d'(s)| \sum_{s'} \mathbb{P}_h^{\pi_h}(s' \mid s) \\
&= \sum_s |d(s) - d'(s)| = \|d - d'\|_1.
\end{aligned}
$$

This finishes the proof. ∎

Now, we are ready to prove Lemma 4.

*Proof of Lemma 4.* For any policy $\pi$ and $\pi'$ and any initial state $s_1$, denote by $d_h$ the distribution of the MDP in step $h$ under policy $\pi_h$ and $d'_h$ the distribution of the MDP in step $h$ under policy $\pi'_h$, that is $d_h(s_h) = \mathbb{E}_\pi[\mathbb{1}(s_h) \mid s_1]$ and $d'_h(s_h) = \mathbb{E}_{\pi'}[\mathbb{1}(s_h) \mid s_1]$. We have

$$
\begin{aligned}
\left\| d_h - d'_h \right\|_1 &= \| d_{h-1}\mathbb{P}_h^{\pi_h} - d'_{h-1}\mathbb{P}_h^{\pi'_h} \|_1 \\
&\le \left\| d_{h-1}\mathbb{P}_h^{\pi_h} - d'_{h-1}\mathbb{P}_h^{\pi_h} \right\|_1 + \left\| d'_{h-1}\mathbb{P}_h^{\pi_h} - d'_{h-1}\mathbb{P}_h^{\pi'_h} \right\|_1 \\
&\le \left\| d_{h-1} - d'_{h-1} \right\|_1 + \left\| \pi_h - \pi'_h \right\|_{1,\infty} \\
&\le \sum_{i \in [h]} \left\| \pi_i - \pi'_i \right\|_{1,\infty},
\end{aligned}
$$

where the second inequality holds by Lemmas 6 and 7 and the last inequality holds by a recursive calculation. Thus, we have

$$\sum_{k=1}^K \left( \mathbb{E}_{\pi_k^c} - \mathbb{E}_{\pi_{k-1}^c} \right) \left[ \sum_{h=1}^H \mathbb{1}(s_h) \mid s_1 = s_{k,1} \right] \le \sum_{k=1}^K \sum_{h=1}^H \sum_{i=1}^h \|\pi_{k,i}^c - \pi_{k-1,i}^c\|_{1,\infty} = HP_T.$$

This finishes the proof. ∎

### C.5 Proof of Theorem 1

*Proof.* By Lemma 2, we have

$$\text{D-Regret}(K) \le \frac{\eta K H^3}{2} + \frac{1}{\eta} \left( P_T \log \frac{4A}{\gamma} + KH \log \frac{1}{1-\gamma} \right) \tag{19}$$

$$+ \frac{1}{\eta} \sum_{k=1}^K \sum_{h=1}^H \mathbb{E}_{\pi_k^c} \left[ \sum_{a \in \mathcal{A}} \left( \pi_{k-1,h}^c(a \mid s_h) \log \frac{1}{\pi'_{k,h}(a \mid s_h)} - \pi_{k,h}^c(a \mid s_h) \log \frac{1}{\pi'_{k+1,h}(a \mid s_h)} \right) \right]$$

Then, the last term can be bounded as follows:

$$\sum_{k=1}^{K}\sum_{h=1}^{H}\mathbb{E}_{\pi_k^c}\left[\sum_{a\in\mathcal{A}}\left(\pi_{k-1,h}^c(a\mid s_h)\log\frac{1}{\pi_{k,h}'(a\mid s_h)}-\pi_{k,h}^c(a\mid s_h)\log\frac{1}{\pi_{k+1,h}'(a\mid s_h)}\right)\right]$$

$$\leq\sum_{k=1}^{K}\sum_{h=1}^{H}\left(\mathbb{E}_{\pi_{k-1}^c}\left[\sum_{a\in\mathcal{A}}\pi_{k-1,h}^c(a\mid s_h)\log\frac{1}{\pi_{k,h}'(a\mid s_h)}\right]-\mathbb{E}_{\pi_k^c}\left[\sum_{a\in\mathcal{A}}\pi_{k,h}^c(a\mid s_h)\log\frac{1}{\pi_{k+1,h}'(a\mid s_h)}\right]\right)$$

$$+\sum_{k=1}^{K}\sum_{h=1}^{H}\mathbb{E}_{\pi_k^c-\pi_{k-1}^c}\left[\sum_{a\in\mathcal{A}}\pi_{k-1,h}^c(a\mid s_h)\log\frac{1}{\pi_{k,h}'(a\mid s_h)}\right]$$

$$\leq\mathbb{E}_{\pi_1^c}\left[\sum_{a\in\mathcal{A}}\pi_{1,h}^c(a\mid s_h)\log\frac{1}{\pi_{1,h}'(a\mid s_h)}\right]+\sum_{k=1}^{K}\sum_{h=1}^{H}\mathbb{E}_{\pi_k^c-\pi_{k-1}^c}\left[\sum_{a\in\mathcal{A}}\pi_{k-1,h}^c(a\mid s_h)\log\frac{1}{\pi_{k,h}'(a\mid s_h)}\right]$$

$$\leq H\log A+\log\frac{4A}{\gamma}\sum_{k=1}^{K}\sum_{h=1}^{H}\mathbb{E}_{\pi_k^c-\pi_{k-1}^c}\left[\mathbb{1}(s_h)\mid s_1=s_{k,1}\right]$$

$$\leq H\log A+\log\frac{4A}{\gamma}HP_T, \tag{20}$$

where the second inequality holds by the telescoping sum and the third inequality holds because $\pi_{1,h}'=\pi_{1,h}$ is the uniform distribution and $\pi_{k,h}'(a\mid s)\geq\frac{\gamma}{4A}$ by the multiplicative stability lemma in Lemma 3 and the last inequality holds by Lemma 4.

Combining (19) and (20), we have

$$\text{D-Regret}(K)\leq\frac{\eta KH^3}{2}+\frac{1}{\eta}\left(H\log A+(1+H)\log\frac{4A}{\gamma}P_T+KH\log\frac{1}{1-\gamma}\right),$$

which finishes the proof. ∎

### C.6 Proof of Theorem 2

To prove Theorem 2, we first introduce the following key lemma omitted in the main paper due to the space limit, which shows the guarantee of the confidence set.

**Lemma 8** (Zhou et al. [31, Lemma 5]). *Let $\widehat{\mathcal{C}}_{k,h}$ be defined in (10) and set $\widehat{\beta}_k$ as*

$$\widehat{\beta}_k=8\sqrt{d\log(1+k/\lambda)\log\left(4k^2H/\delta\right)}+4\sqrt{d}\log\left(4k^2H/\delta\right)+\sqrt{\lambda}B.$$

*Then, with probability at least $1-3\delta$, we have that simultaneously for all $h\in[H]$ and $k\in[K]$,*

$$\theta_h^*\in\left\{\theta\Big|\big\|\widehat{\Sigma}_{k,h}^{1/2}(\theta-\widehat{\theta}_{k,h})\big\|_2\leq\widehat{\beta}_k\right\},\left|\left[\bar{\mathbb{V}}_{k,h}V_{k,h+1}\right](s_{k,h},a_{k,h})-\left[\mathbb{V}_hV_{k,h+1}\right](s_{k,h},a_{k,h})\right|\leq E_{k,h},$$

*where $E_{k,h}$ is defined as follows:*

$$\min\left\{H^2,2H\breve{\beta}_k\left\|\widehat{\Sigma}_{k,h}^{-1/2}\phi_{V_{k,h+1}}(s_{k,h},a_{k,h})\right\|_2\right\}+\min\left\{H^2,\widetilde{\beta}_k\left\|\widetilde{\Sigma}_{k,h}^{-1/2}\phi_{V_{k,h+1}^2}(s_{k,h},a_{k,h})\right\|_2\right\}$$

*with*

$$\breve{\beta}_k=8d\sqrt{\log(1+k/\lambda)\log\left(4k^2H/\delta\right)}+4\sqrt{d}\log\left(4k^2H/\delta\right)+\sqrt{\lambda}B,$$

$$\widetilde{\beta}_k=8\sqrt{dH^4\log\left(1+kH^4/(d\lambda)\right)\log\left(4k^2H/\delta\right)}+4H^2\log\left(4k^2H/\delta\right)+\sqrt{\lambda}B. \tag{21}$$

We denote by $\mathcal{E}$ the event when the result of Lemma 8 holds, we have $\Pr(\mathcal{E})\geq 1-3\delta$.

Then we present the following lemma that shows the value function difference can be decomposed into two parts, one is a martingale sequence and the other is the estimation error.

**Lemma 9.** *For all $k\in[K],h\in[H]$, it holds that*

$$V_{k,h}(s_{k,h})-V_{k,h}^{\pi_k}(s_{k,h})=\sum_{h'=h}^{H}\left(M_{k,h',1}+M_{k,h',2}-\iota_{k,h'}(s_{k,h'},a_{k,h'})\right),$$

*with*

$$M_{k,h,1} = \mathbb{E}_{a \sim \pi_k(\cdot \mid s_{k,h})}[Q_{k,h}(s_{k,h}, a) - Q_{k,h}^{\pi_k}(s_{k,h}, a)] - (Q_{k,h}(s_{k,h}, a_{k,h}) - Q_{k,h}^{\pi_k}(s_{k,h}, a_{k,h})),$$

$$M_{k,h,2} = [\mathbb{P}_h(V_{k,h+1} - V_{k,h+1}^{\pi_k})](s_{k,h}, a_{k,h}) - (V_{k,h+1}(s_{k,h+1}) - V_{k,h+1}^{\pi_k}(s_{k,h+1})).$$

*Proof.* By the definition $V_{k,h}(s_{k,h}) = \mathbb{E}_{a \sim \pi_k(\cdot \mid s_{k,h})}[Q_{k,h}(s_{k,h}, a)]$, we have

$$
\begin{aligned}
& V_{k,h}(s_{k,h}) - V_{k,h}^{\pi_k}(s_{k,h}) \\
&= \mathbb{E}_{a \sim \pi_k(\cdot \mid s_{k,h})}\big[Q_{k,h}(s_{k,h}, a) - Q_{k,h}^{\pi_k}(s_{k,h}, a)\big] \\
&= \mathbb{E}_{a \sim \pi_k(\cdot \mid s_{k,h})}\big[Q_{k,h}(s_{k,h}, a) - Q_{k,h}^{\pi_k}(s_{k,h}, a)\big] - \big(Q_{k,h}(s_{k,h}, a_{k,h}) - Q_{k,h}^{\pi_k}(s_{k,h}, a_{k,h})\big) \\
&\quad + \big(Q_{k,h}(s_{k,h}, a_{k,h}) - Q_{k,h}^{\pi_k}(s_{k,h}, a_{k,h})\big) \\
&= \mathbb{E}_{a \sim \pi_k(\cdot \mid s_{k,h})}\big[Q_{k,h}(s_{k,h}, a) - Q_{k,h}^{\pi_k}(s_{k,h}, a)\big] - \big(Q_{k,h}(s_{k,h}, a_{k,h}) - Q_{k,h}^{\pi_k}(s_{k,h}, a_{k,h})\big) \\
&\quad + [\mathbb{P}_h(V_{k,h+1} - V_{k,h+1}^{\pi_k})](s_{k,h}, a_{k,h}) - \iota_{k,h}(s_{k,h}, a_{k,h}) \\
&= \underbrace{\mathbb{E}_{a \sim \pi_k(\cdot \mid s_{k,h})}\big[Q_{k,h}(s_{k,h}, a) - Q_{k,h}^{\pi_k}(s_{k,h}, a)\big] - \big(Q_{k,h}(s_{k,h}, a_{k,h}) - Q_{k,h}^{\pi_k}(s_{k,h}, a_{k,h})\big)}_{M_{k,h,1}} \\
&\quad + \underbrace{[\mathbb{P}_h(V_{k,h+1} - V_{k,h+1}^{\pi_k})](s_{k,h}, a_{k,h}) - \big(V_{k,h+1}(s_{k,h+1}) - V_{k,h+1}^{\pi_k}(s_{k,h+1})\big)}_{M_{k,h,2}} \\
&\quad + \big(V_{k,h+1}(s_{k,h+1}) - V_{k,h+1}^{\pi_k}(s_{k,h+1})\big) - \iota_{k,h}(s_{k,h}, a_{k,h})
\end{aligned}
$$

where the third equality holds by the fact $Q_{k,h} = r_{k,h} + \mathbb{P}_h V_{k,h+1} - \iota_{k,h}$ and $Q_{k,h}^{\pi_k} = r_{k,h} + \mathbb{P}_h V_{k,h+1}^{\pi_k}$. Summing up the above equation from $h$ to $H$ recursively finishes the proof. ∎

Note $M_{k,h,1}$ is the noise from the stochastic policy and $M_{k,h,2}$ is the noise from the state transition, Let $M_{k,h} = M_{k,h,1} + M_{k,h,2}, \forall k \in [K], h \in [H]$, we define two following high probability events:

$$\mathcal{E}_1 = \bigg\{ \forall h \in [H], \sum_{k=1}^{K} \sum_{h'=h}^{H} M_{k,h'} \leq 4\sqrt{H^3 K \log \frac{H}{\delta}} \bigg\}, \mathcal{E}_2 = \bigg\{ \sum_{k=1}^{K} \sum_{h=1}^{H} M_{k,h,2} \leq \sqrt{8H^3 K \log \frac{1}{\delta}} \bigg\}.$$

According to the Azuma-Hoeffding inequality, we have $\Pr(\mathcal{E}_1) \geq 1 - \delta$ and $\Pr(\mathcal{E}_2) \geq 1 - \delta$.

Then, we show the model prediction error $\iota_{k,h}$ can be upper and lower bounded both.

**Lemma 10.** *Define prediction error $\iota_{k,h} = r_{k,h} + \mathbb{P}_h V_{k,h+1} - Q_{k,h}$, on the event $\mathcal{E}$, it holds that*

$$-2\widehat{\beta}_k \big\| \widehat{\Sigma}_{k,h}^{-1/2} \phi_{V_{k,h+1}}(\cdot, \cdot) \big\|_2 \leq \iota_{k,h}(\cdot, \cdot) \leq 0, \forall k \in [K], h \in [H].$$

*Proof.* First, we prove the left-hand side inequality. By the definition of $\iota_{k,h}$, we have

$$
\begin{aligned}
& -\iota_{k,h}(s, a) \\
&= Q_{k,h}(s, a) - (r_{k,h} + \mathbb{P}_h V_{k,h+1})(s, a) \\
&\leq r_{k,h}(s, a) + \big\langle \widehat{\theta}_{k,h}, \phi_{V_{k,h+1}}(s, a) \big\rangle + \widehat{\beta}_k \big\| \widehat{\Sigma}_{k,h}^{-1/2} \phi_{V_{k,h+1}}(s, a) \big\|_2 - (r_{k,h} + \mathbb{P}_h V_{k,h+1})(s, a) \\
&= \big\langle \widehat{\theta}_{k,h} - \theta_h^*, \phi_{V_{k,h+1}}(s, a) \big\rangle + \widehat{\beta}_k \big\| \widehat{\Sigma}_{k,h}^{-1/2} \phi_{V_{k,h+1}}(s, a) \big\|_2 \\
&\leq 2\widehat{\beta}_k \big\| \widehat{\Sigma}_{k,h}^{-1/2} \phi_{V_{k,h+1}}(s, a) \big\|_2,
\end{aligned}
$$

where the first inequality holds by the configuration of $Q_{k,h}$ in Algorithm 1, the second equality holds by the definition of linear mixture MDP such that $[\mathbb{P}_h V_{k,h+1}](s, a) = \langle \phi_{V_{k,h+1}}(s, a), \theta_h^* \rangle$ and the last inequality holds by the configuration of confidence set in Lemma 8.

Next, we prove the right-hand side inequality. By the definition of $\iota_{k,h}$, we have

$$
\begin{aligned}
& \iota_{k,h}(s, a) \\
&= (r_{k,h} + \mathbb{P}_h V_{k,h+1})(s, a) - Q_{k,h}(s, a) \\
&\leq (r_{k,h} + \mathbb{P}_h V_{k,h+1})(s, a) - \Big[ r_{k,h}(s, a) + \big\langle \widehat{\theta}_{k,h}, \phi_{V_{k,h+1}}(s, a) \big\rangle + \widehat{\beta}_k \big\| \widehat{\Sigma}_{k,h}^{-1/2} \phi_{V_{k,h+1}}(s, a) \big\|_2 \Big]_{[0, H-h+1]}
\end{aligned}
$$

$$= \max\left\{\langle\widehat{\theta}_{k,h} - \theta_h^*, \phi_{V_{k,h+1}}(s,a)\rangle - \widehat{\beta}_k \big\|\widehat{\Sigma}_{k,h}^{-1/2}\phi_{V_{k,h+1}}(s,a)\big\|_2, (r_{k,h} + \mathbb{P}_h V_{k,h+1})(s,a) - (H-h+1)\right\}$$
$$\leq 0,$$

where the first inequality holds by the configuration of $Q_{k,h}$ in Algorithm 1, the second equality holds by the definition of linear mixture MDP such that $[\mathbb{P}_h V_{k,h+1}](s,a) = \langle\phi_{V_{k,h+1}}(s,a), \theta_h^*\rangle$ and the last inequality holds by the configuration of confidence set in Lemma 8. ∎

Next, we show the prediction error can be bounded by the cumulative estimate variance.

**Lemma 11.** *Define prediction error* $\iota_{k,h} = r_{k,h} + \mathbb{P}_h V_{k,h+1} - Q_{k,h}$, *on the event* $\mathcal{E}$, *it holds that*

$$-\sum_{k=1}^{K}\sum_{h=1}^{H} \iota_{k,h}(s_{k,h}, a_{k,h}) \leq 2\widehat{\beta}_K\sqrt{\sum_{k=1}^{K}\sum_{h=1}^{H}\bar{\sigma}_{k,h}^2}\sqrt{2Hd\log(1+K/\lambda)}.$$

*Proof.* By Lemma 10 and the definition of $\iota_{k,h} = r_{k,h} + \mathbb{P}_h V_{k,h+1} - Q_{k,h}$, we have

$$-\sum_{k=1}^{K}\sum_{h=1}^{H}\iota_{k,h}(s_{k,h}, a_{k,h})$$
$$\leq \sum_{k=1}^{K}\sum_{h=1}^{H} 2\min\{\widehat{\beta}_k\big\|\widehat{\Sigma}_{k,h}^{-1/2}\phi_{V_{k,h+1}}(s_{k,h}, a_{k,h})\big\|_2, H\}$$
$$\leq \sum_{k=1}^{K}\sum_{h=1}^{H} 2\widehat{\beta}_k\bar{\sigma}_{k,h}\min\left\{\big\|\widehat{\Sigma}_{k,h}^{-1/2}\phi_{V_{k,h+1}}(s_{k,h}, a_{k,h})/\bar{\sigma}_{k,h}\big\|_2, 1\right\}$$
$$\leq 2\widehat{\beta}_K\sqrt{\sum_{k=1}^{K}\sum_{h=1}^{H}\bar{\sigma}_{k,h}^2}\sqrt{\sum_{k=1}^{K}\sum_{h=1}^{H}\min\left\{\big\|\widehat{\Sigma}_{k,h}^{-1/2}\phi_{V_{k,h+1}}(s_{k,h}, a_{k,h})/\bar{\sigma}_{k,h}\big\|_2, 1\right\}}$$
$$\leq 2\widehat{\beta}_K\sqrt{\sum_{k=1}^{K}\sum_{h=1}^{H}\bar{\sigma}_{k,h}^2}\sqrt{2Hd\log(1+K/\lambda)},$$

where the second inequality holds by $2\widehat{\beta}_k\bar{\sigma}_{k,h} \geq \sqrt{d}H/\sqrt{d} = H$, the third inequality is by Cauchy-Schwarz inequality and the last inequality holds by the elliptical potential lemma in Lemma 16. ∎

Define the event

$$\mathcal{E}_3 = \left\{\sum_{k=1}^{K}\sum_{h=1}^{H}[\mathbb{V}_h V_{h+1}^{\pi^k}](s_h^k, a_h^k) \leq 3(HK + H^3\log(1/\delta))\right\},$$

by the law of total variance in Lemma 15, we have $\Pr(\mathcal{E}_3) \geq 1 - \delta$. Then, we have the following lemma which bounds the estimated variance of the value function.

**Lemma 12** (He et al. [33, Lemma 6.5]). *On the events* $\mathcal{E} \cap \mathcal{E}_1 \cap \mathcal{E}_2 \cap \mathcal{E}_3$, *it holds that*

$$\sum_{k=1}^{K}\sum_{h=1}^{H}\bar{\sigma}_{k,h}^2 \leq 2H^3K/d + 179H^2K + 165d^3H^4\log^2\left(4K^2H/\delta\right)\log^2\left(1+KH^4/\lambda\right)$$
$$+ 2062d^2H^5\log^2\left(4K^2H/\delta\right)\log^2(1+K/\lambda).$$

Now, we are ready to prove Theorem 2.

*Proof of Theorem 2.* On the events $\mathcal{E} \cap \mathcal{E}_1 \cap \mathcal{E}_2 \cap \mathcal{E}_3$, for any $h \in [H]$, it holds that

$$\sum_{k=1}^{K} V_{k,h}(s_{k,h}) - \sum_{k=1}^{K} V_{k,h}^{\pi_k}(s_{k,h})$$

$$\leq \sum_{k=1}^{K} \sum_{h=1}^{H} (M_{k,h,1} + M_{k,h,2} - \iota_{k,h}(s_{k,h}, a_{k,h}))$$

$$\leq 4\sqrt{H^3 K \log(H/\delta)} + 2\widehat{\beta}_K \sqrt{\sum_{k=1}^{K}\sum_{h=1}^{H} \bar{\sigma}_{k,h}^2} \sqrt{2Hd\log(1 + K/\lambda)}$$

$$\leq \widetilde{\mathcal{O}}\big(\sqrt{dH^4 K + d^2 H^3 K}\big),$$

where the first equality holds by Lemma 9, the second inequality holds by Lemma 11, and the last inequality holds by Lemma 12. This finishes the proof. ∎

### C.7 Proof of Theorem 3

*Proof.* By Lemma 5, we can rewrite the dynamic regret as follows.

$$\begin{aligned}
\text{D-Regret}(K) &= \sum_{k=1}^{K} V_{k,1}^{\pi_k^c}(s_{k,1}) - \sum_{k=1}^{K} V_{k,1}^{\pi_k}(s_{k,1}) \\
&= \sum_{k=1}^{K}\sum_{h=1}^{H} \mathbb{E}_{\pi_k^c}[\langle Q_{k,h}(s_h, \cdot), \pi_{k,h}^c(\cdot \mid s_h) - \pi_{k,h}(\cdot \mid s_h)\rangle] \\
&\quad + \sum_{k=1}^{K}\sum_{h=1}^{H} (\mathbb{E}_{\pi_k^c}[\iota_{k,h}(s_h, a_h)] - \iota_{k,h}(s_{k,h}, a_{k,h})) + \mathcal{M}_{K,H}.
\end{aligned} \tag{22}$$

By Lemma 10, we have $\iota_{k,h}(s, a) \leq 0$ for any $k \in [K], h \in [H], s \in \mathcal{S}, a \in \mathcal{A}$. Thus, we have

$$\sum_{k=1}^{K}\sum_{h=1}^{H} \mathbb{E}_{\pi_k^c}[\iota_{k,h}(s_h, a_h)] \leq 0. \tag{23}$$

By Theorem 2, we have

$$\sum_{k=1}^{K}\sum_{h=1}^{H}(-\iota_{k,h}(s_{k,h}, a_{k,h})) + \mathcal{M}_{K,H} \leq \widetilde{\mathcal{O}}\big(\sqrt{dH^4 K + d^2 H^3 K}\big). \tag{24}$$

It remains to bound the first term. Note our algorithm is indeed updated based on the estimated action-value function $Q_{k,h}$. By Theorem 1, set $\gamma = 1/KH$ and note that $\log(1/(1-\gamma)) \leq \gamma/(1-\gamma)$ for all $\gamma > 0$. Then, we have

$$\begin{aligned}
&\sum_{k=1}^{K}\sum_{h=1}^{H} \mathbb{E}_{\pi_k^c}[\langle Q_{k,h}(s_h, \cdot), \pi_{k,h}^c(\cdot \mid s_h) - \pi_{k,h}(\cdot \mid s_h)\rangle] \\
&\leq \frac{\eta K H^3}{2} + \frac{1}{\eta}\left(H \log A + (1 + H)\log\frac{A}{\gamma}P_T + 2\right)
\end{aligned}$$

It is clear that the optimal step size is $\eta^* = \sqrt{(P_T + \log A)/(KH^2)}$. Note our step size is set as $\eta = \min\{\sqrt{(P_T + \log A)/K}, 1\}/H$ to ensure $\eta \leq 1/H$. We consider the following two cases:

**Case 1:** $\eta^* \leq 1/H$. In this case, our step size is set as $\eta = \sqrt{(P_T + \log A)/(KH^2)}$. We have

$$\frac{\eta K H^3}{2} + \frac{1}{\eta}\left(H \log A + (1 + H)\log\frac{A}{\gamma}P_T + 2\right) \leq \widetilde{\mathcal{O}}(\sqrt{KH^4(1 + P_T)}).$$

**Case 2:** $\eta^* > 1/H$. In this case, our step size is set as $\eta = 1/H$. Therefore, we have

$$\frac{\eta K H^3}{2} + \frac{1}{\eta}\left(H \log A + (1 + H)\log\frac{A}{\gamma}P_T + 2\right) \leq \widetilde{\mathcal{O}}(H^2 P_T),$$

where the equality holds by $P_T \geq K$ in this case. Combining these two cases, we have

$$\sum_{k=1}^{K}\sum_{h=1}^{H} \mathbb{E}_{\pi_k^c}[\langle Q_{k,h}(s_h,\cdot), \pi_{k,h}^c(\cdot \mid s_h) - \pi_{k,h}(\cdot \mid s_h)\rangle] \leq \widetilde{\mathcal{O}}(\sqrt{H^4(K+P_T)(1+P_T)}). \quad (25)$$

Combining (22), (23), (24) and (25), we obtain

$$\text{D-Regret}(K) \leq \widetilde{\mathcal{O}}\left(\sqrt{dH^4K + d^2H^3K} + \sqrt{H^4(K+P_T)(1+P_T)}\right).$$

This finishes the proof. ∎

### C.8 Proof of Theorem 4

*Proof.* At a high level, we prove this lower bound by noting that optimizing the dynamic regret of linear mixture MDPs is harder than (i) optimizing the static regret of linear mixture MDPs with the unknown transition kernel, (ii) optimizing the dynamic regret of linear mixture MDPs with the known transition kernel, both. Thus, we can consider the lower bound of these two problems separately and combine them to obtain the lower bound of the dynamic regret of linear mixture MDPs with the unknown transition kernel. We present the details below.

First, we consider the lower bound of optimizing the static regret of adversarial linear mixture MDPs with the unknown transition kernel. From lower bound in He et al. [33, Theorem 5.3], we have the following lower bound in this case since the dynamic regret is no smaller than the static regret.

$$\text{D-Regret}(K) \geq \Omega(\sqrt{d^2H^3K}). \quad (26)$$

Then, we consider the lower bound of optimizing the dynamic regret of adversarial linear mixture MDPs with the known transition kernel. We note that Zimin and Neu [12] show the lower bound of the static regret for adversarial episodic loop-free SSP with known transition kernel is $\Omega(H\sqrt{K\log(SA)})$, we utilize this lower bound to establish our lower bound as the episodic loop-free SSP is a special case of linear mixture MDPs with $d = S^2A$. We consider two cases:

**Case 1:** $\Gamma \leq 2H$. In this case, we can directly utilize the established lower bound of static regret as a natural lower bound of dynamic regret,

$$\text{D-Regret}(K) \geq \Omega(H\sqrt{K\log(SA)}). \quad (27)$$

**Case 2:** $\Gamma > 2H$. Without loss of generality, we assume $L = \lceil \Gamma/2H \rceil$ divides $K$ and split the whole episodes into $L$ pieces equally. Next, we construct a special policy sequence such that the policy sequence is fixed within each piece and only changes in the split point. Since the sequence changes at most $L - 1 \leq \Gamma/2H$ times and the path length of the policy sequence at each change point is at most $2H$, the total path length in $K$ episodes does not exceed $\Gamma$. As a result, we have

$$\text{D-Regret}(K) \geq LH\sqrt{K/L\log(SA)} = H\sqrt{KL\log(SA)} \geq \Omega(\sqrt{KH\Gamma\log(SA)}). \quad (28)$$

Combining (27) and (28), we have the following lower bound for the dynamic regret of adversarial linear mixture MDPs with the known transition kernel,

$$\text{D-Regret}(K) \geq \Omega\left(\max\{H\sqrt{K\log(SA)}, \sqrt{KH\Gamma\log(SA)}\}\right) \geq \Omega(\sqrt{KH(H+\Gamma)\log(SA)}), \quad (29)$$

where the last inequality holds by $\max\{a,b\} \geq (a+b)/2$.

Combining two lower bounds (26) and (29), we have the lower bound of the dynamic regret of adversarial linear mixture MDPs with the unknown transition kernel,

$$\begin{aligned}
\text{D-Regret}(K) &\geq \Omega\left(\max\{\sqrt{d^2H^3K}, \sqrt{KH(H+\Gamma)\log(SA)}\}\right) \\
&\geq \Omega\left(\sqrt{d^2H^3K} + \sqrt{KH(H+\Gamma)\log(SA)}\right).
\end{aligned}$$

This finishes the proof. ∎

## C.9  Proof of Theorem 5

*Proof.* By Lemma 5, we can rewrite the dynamic regret as follows.

$$\text{D-Regret}(K) = \sum_{k=1}^{K} V_{k,1}^{\pi_k^c}(s_{k,1}) - \sum_{k=1}^{K} V_{k,1}^{\pi_k}(s_{k,1})$$

$$= \sum_{k=1}^{K} \sum_{h=1}^{H} \mathbb{E}_{\pi_k^c}[\langle Q_{k,h}(s_h, \cdot), \pi_{k,h}^c(\cdot \mid s_h) - \pi_{k,h}(\cdot \mid s_h) \rangle]$$

$$+ \sum_{k=1}^{K} \sum_{h=1}^{H} (\mathbb{E}_{\pi_k^c}[\iota_{k,h}(s_h, a_h)] - \iota_{k,h}(s_{k,h}, a_{k,h})) + \mathcal{M}_{K,H}.$$

By Lemma 10, we have $\iota_{k,h}(s, a) \leq 0$ for any $k \in [K], h \in [H], s \in \mathcal{S}, a \in \mathcal{A}$. Thus, we have

$$\sum_{k=1}^{K} \sum_{h=1}^{H} \mathbb{E}_{\pi_k^c}[\iota_{k,h}(s_h, a_h)] \leq 0.$$

By Theorem 2, we have

$$\sum_{k=1}^{K} \sum_{h=1}^{H} (-\iota_{k,h}(s_{k,h}, a_{k,h})) + \mathcal{M}_{K,H} \leq \widetilde{\mathcal{O}}\big(\sqrt{dH^4 K + d^2 H^3 K}\big).$$

It remains to bound the first term. We decompose this term as follows.

$$\sum_{k=1}^{K} \sum_{h=1}^{H} \mathbb{E}_{\pi_k^c} \left[ \langle Q_{k,h}^{\pi_k}(s_h, \cdot), \pi_{k,h}^c(\cdot \mid s_h) - \pi_{k,h}(\cdot \mid s_h) \rangle \right]$$

$$= \underbrace{\sum_{k=1}^{K} \sum_{h=1}^{H} \mathbb{E}_{\pi_k^c} \left[ \langle Q_{k,h}^{\pi_k}(s_h, \cdot), \pi_{k,h}^c(\cdot \mid s_h) - \pi_{k,h}^i(\cdot \mid s_h) \rangle \right]}_{\texttt{base-regret}}$$

$$+ \underbrace{\sum_{k=1}^{K} \sum_{h=1}^{H} \mathbb{E}_{\pi_k^c} \left[ \langle Q_{k,h}^{\pi_k}(s_h, \cdot), \pi_{k,h}^i(\cdot \mid s_h) - \pi_{k,h}(\cdot \mid s_h) \rangle \right]}_{\texttt{meta-regret}}, \quad (30)$$

where the decomposition holds for any base-learner $i \in N$. Next, we bound the two terms separately.

**Upper bound of base regret.**  From Theorem 1, we have

$$\texttt{base-regret} \leq \frac{\eta_i K H^3}{2} + \frac{1}{\eta_i} \left( H \log A + (1 + H) \log \frac{A}{\gamma} P_T + K H \log \frac{1}{1 - \gamma} \right)$$

Set $\gamma = 1/KH$ and note that $\log(1/(1 - \gamma)) \leq \gamma/(1 - \gamma)$ for all $\gamma > 0$. Then, we have

$$\texttt{base-regret} \leq \frac{\eta_i K H^3}{2} + \frac{1}{\eta_i} \left( H \log A + (1 + H) P_T \log(KHA) + 2 \right)$$

$$\leq \frac{\eta_i K H^3}{2} + \frac{2H}{\eta_i} \left( \log A + P_T \log(KHA) \right).$$

It is clear that the optimal learning rate $\eta^* = \sqrt{4(\log A + P_T \log(KHA))/(KH^2)}$. By the definition $P_T = \sum_{k=1}^{K} \sum_{h=1}^{H} \|\pi_{k,h}^c - \pi_{k-1,h}^c\|_{1,\infty}$, it holds that $0 \leq P_T \leq 2KH$. Therefore, the range of the optimal learning rate is

$$\eta_{\min} = \sqrt{\frac{4 \log A}{KH^2}}, \text{ and } \eta_{\max} = \sqrt{\frac{4 \log A + 8KH \log(KHA)}{KH^2}}.$$

From the construction of the step size pool $\mathcal{H} = \{\eta_i = (2^i/H)\sqrt{(\log A)/K} \mid i \in [N]\}$ with $N = \lfloor \frac{1}{2} \log(\frac{K}{\log A}) \rfloor$, we know that the step size therein is monotonically increasing, in particular

$$\eta_1 = \sqrt{\frac{4 \log A}{KH^2}}, \text{ and } \eta_N = \frac{1}{H}.$$

In the following, we consider two cases:

**Case 1: $\eta^* \in [\eta_1, \eta_N]$.** In this case, we ensure there exists $i^* \in N$ such that $\eta_{i^*} \leq \eta^* \leq 2\eta_{i^*}$. Note the decomposition in (30) holds for any base-learner. Therefore, we choose the base-learner whose step size is $\eta_{i^*}$ and have

$$
\begin{aligned}
\texttt{base-regret} &\leq \frac{\eta_{i^*} KH^3}{2} + \frac{2H}{\eta_{i^*}} \left(\log A + P_T \log(KHA)\right) \\
&\leq \frac{\eta^* KH^3}{2} + \frac{4H}{\eta^*} \left(\log A + P_T \log(KHA)\right) \\
&= 3\sqrt{KH^4 \left(\log A + P_T \log(KHA)\right)},
\end{aligned}
$$

where the second inequality holds by the condition that $\eta_{i^*} \leq \eta^* \leq 2\eta_{i^*}$ and the last equality holds by substituting $\eta^* = \sqrt{4(\log A + P_T \log(KHA))/(KH^2)}$.

**Case 2: $\eta^* > \eta_N$.** In this case, we know that $4(\log A + P_T \log(KHA)) > K$. Therefore, we choose the base-learner whose step size is $\eta_N$ and have

$$
\begin{aligned}
\texttt{base-regret} &\leq \frac{\eta_N KH^3}{2} + \frac{2H}{\eta_N} \left(\log A + P_T \log(KHA)\right) \\
&= \frac{KH^2}{2} + 2H^2 \left(\log A + P_T \log(KHA)\right) \\
&\leq 4H^2 \left(\log A + P_T \log(KHA)\right),
\end{aligned}
$$

where the last inequality holds by the condition that $4(\log A + P_T \log(KHA)) > K$.

Summing over the two upper bounds yields

$$
\begin{aligned}
\texttt{base-regret} &\leq 3\sqrt{KH^4 \left(\log A + P_T \log(KHA)\right)} + 4H^2 \left(\log A + P_T \log(KHA)\right) \\
&\leq 4\sqrt{H^4(K + \log A + P_T \log(KHA))(\log A + P_T \log(KHA))} \\
&= \widetilde{\mathcal{O}}(\sqrt{H^4(K + P_T)(1 + P_T)}), \quad\quad\quad (31)
\end{aligned}
$$

where the inequality holds by $\sqrt{a} + \sqrt{b} \leq \sqrt{2(a+b)}, \forall a, b \geq 0$.

**Upper bound of meta regret.** For meta-regret, we have

$$
\begin{aligned}
\texttt{meta-regret} &= \sum_{k=1}^{K} \sum_{h=1}^{H} \mathbb{E}_{\pi_k^c} \left[ \langle Q_{k,h}(s_h, \cdot), \pi_{k,h}^i(\cdot \mid s_h) - \pi_{k,h}(\cdot \mid s_h) \rangle \right] \\
&= \sum_{k=1}^{K} \sum_{h=1}^{H} \mathbb{E}_{\pi_k^c} \left[ \langle e^i(s_h) - p_{k,h}^i(s_h), Q_{k,h}(s_h, \cdot) \cdot \pi_{k,h}^i(\cdot \mid s_h) \rangle \right] \\
&\leq \sum_{k=1}^{K} \sum_{h=1}^{H} \mathbb{E}_{\pi_k^c} \left[ \langle e^{i_{k,h}^*}(s_h) - p_{k,h}^i(s_h), Q_{k,h}(s_h, \cdot) \cdot \pi_{k,h}^i(\cdot \mid s_h) \rangle \right] \\
&\leq \sum_{k=1}^{K} \sum_{h=1}^{H} \mathbb{E}_{\pi_k^c} \left[ \langle e^{i_{k,h}^*}(s_h) - e^{i_{k-1,h}^*}(s_h), Q_{k,h}(s_h, \cdot) \cdot \pi_{k,h}^i(\cdot \mid s_h) \rangle \right] \\
&\leq H \sum_{k=1}^{K} \sum_{h=1}^{H} \mathbb{E}_{\pi_k^c} \left[ \|e^{i_{k,h}^*}(s_h) - e^{i_{k-1,h}^*}(s_h)\|_1 \right] \\
&\leq 2HS_T, \quad\quad\quad (32)
\end{aligned}
$$

where the first inequality holds by the definition that $i_{k,h}^* = \arg\max_{i \in [N]} \langle Q_{k,h}(s_h, \cdot), \pi_{k,h}^i(\cdot \mid s_h) \rangle$, the second inequality holds due to $p_{k,h}^i(s_h) = e^{i_{k-1,h}^*}(s_h)$, and the last equality holds by the definition $S_T = \sum_{k=1}^{K} \sum_{h=1}^{H} \mathbb{E}_{\pi_k^c} \mathbb{1}[i_{k,h}^*(s_h) \neq i_{k-1,h}^*(s_h)]$.

Combining (31) and (32), by $\sqrt{a} + \sqrt{b} \leq \sqrt{2(a+b)}, \forall a, b \geq 0$, we have

$$
\text{D-Regret}(K) \leq \widetilde{\mathcal{O}}\Big( \sqrt{dH^4 K + d^2 H^3 K} + \sqrt{H^4(K + P_T)(1 + P_T) + H^2 S_T^2} \Big).
$$

This finishes the proof. ∎

# D Supporting Lemmas

In this section, we introduce the supporting lemmas used in the proofs.

First, we introduce the performance difference lemma which connects the difference between two policies to the difference between their expected total rewards through the Q-function.

**Lemma 13** (Cai et al. [32, Lemma 3.2]). *For any policies $\pi, \pi' \in \Delta(\mathcal{A} \mid \mathcal{S}, H)$, it holds that*

$$
V_{k,1}^{\pi'}(s_{k,1}) - V_{k,1}^{\pi}(s_{k,1}) = \mathbb{E}_{\pi'} \left[ \sum_{h=1}^{H} \langle Q_{k,h}^{\pi}(s_h, \cdot), \pi'_h(\cdot \mid s_h) - \pi_h(\cdot \mid s_h) \rangle \,\Big|\, s_1 = s_{k,1} \right].
$$

Then, we introduce the following lemmas which show the "one-step descent" guarantee.

**Lemma 14** (Cai et al. [32, Lemma 3.3]). *For any distributions $p^*, p \in \Delta(\mathcal{A})$, state $s \in \mathcal{S}$, and function $Q : \mathcal{S} \times \mathcal{A} \to [0, H]$, it holds for $p' \in \Delta(\mathcal{A})$ with $p'(\cdot) \propto p(\cdot) \cdot \exp(\eta \cdot Q(s, \cdot))$ that*

$$
\langle Q(s, \cdot), p^*(\cdot) - p(\cdot) \rangle \le \eta H^2 / 2 + \eta^{-1} \cdot \left( D_{\mathrm{KL}}\left(p^*(\cdot) \| p(\cdot)\right) - D_{\mathrm{KL}}\left(p^*(\cdot) \| p'(\cdot)\right) \right).
$$

Next, we introduce the law of total variance, which bounds the variance of the value function.

**Lemma 15** (Jin et al. [55, Lemma C.5]). *With probability at least $1 - \delta$, it holds that*

$$
\sum_{k=1}^{K} \sum_{h=1}^{H} \left[ \mathbb{V}_h V_{h+1}^{\pi^k} \right](s_h^k, a_h^k) \le 3 \left( HK + H^3 \log \frac{1}{\delta} \right).
$$

Finally, we introduce the elliptical potential lemma, which is a key lemma in online linear regression.

**Lemma 16** (Abbasi-Yadkori et al. [51, Lemma 11]). *Let $\{\mathbf{x}_t\}_{t=1}^{\infty}$ be a sequence in $\mathbb{R}^d$ space, $\mathbf{V}_0 = \lambda \mathbf{I}$ and define $\mathbf{V}_t = \mathbf{V}_0 + \sum_{i=1}^{t} \mathbf{x}_i \mathbf{x}_i^\top$. If $\|\mathbf{x}_i\|_2 \le L, \forall i \in \mathbb{Z}_+$, then for each $t \in \mathbb{Z}_+$,*

$$
\sum_{i=1}^{t} \min \left\{ 1, \|\mathbf{x}_i\|_{\mathbf{V}_{i-1}^{-1}} \right\} \le 2d \log \left( \frac{d\lambda + tL^2}{d\lambda} \right).
$$

