# OpenReview forum: "Dynamic Regret of Adversarial Linear Mixture MDPs"
_NeurIPS.cc/2023/Conference — NeurIPS 2023 poster_

### Official Review · Reviewer_BpVE · 2023-07-04

**Soundness:** 2 fair
**Presentation:** 3 good
**Contribution:** 3 good
**Rating:** 6
**Confidence:** 3

**Summary:**

This paper studiess adversarial RL under episodic linear mixture MDPs with adversarial full-information reward and stationary unknown transition kernels. This paper first proposes a new algorithm to deal with the adversarial reward and then provides an upper bound of dynamic regret for the proposed algorithm. Then this paper proposes a lower bound for dynamic regret and claims acheiving optimal in terms of the number of episodes $K$ and the non-stationary measure $P_T$.

**Strengths:**

1. This paper provides a noval algorithm to deal with adversarial RL without prior knowledge. The main steps of the algorithm seem reasonable and the paper provides detailed insight of each steps.
2. This paper provide an upper bound of dynamic regret and the dependenece of $K$ in upper bound achieves optimal.

**Weaknesses:**

1. The algorithm and result of this paper only apply to finite state case, which makes the result less significant.

2. The proof of the lower bound lacks rigor in several aspects. Firstly, the proof of the lower bound is divided into two separate cases, which is not in line with the definition of a lower bound. Ideally, the author should construct one hard instance that encompasses both cases. Secondly, in the proof of the second case (dynamic regret case), the proof merely takes the maximum between the results of the two cases with respect to the range of $\Gamma$.

3. Due to the existence of numerous studies employing alternative approaches to address non-stationarity and yield diverse results, I think more comprehensive and detialed discussions with non-stationary RL are needed. For example, although line 108 indicates that [1] is not applicable to the adversarial setting when the comparators are arbitrarily chosen. In fact, people are usually concerned about the case where camparators are optimal policy. In this case, does the non-stationary RL discussed in [1] cover the adversarial RL in this paper? It is important to discuss the relationship between the findings of [1] and this paper in such cases. [1] also achieves optimal regret with $K^{2/3}\Delta^{1/3}+\sqrt{K}$, where $\Delta$ denotes another non-stationary measure. Why does the dependence on $K$ differ? From my understanding, this discrepancy seems to arise from distinct non-stationary measures.


[1] Chen-Yu Wei and Haipeng Luo. Non-stationary reinforcement learning without prior knowledge: An optimal black-box approach. In Proceedings of the 34th Conference on Learning Theory (COLT), pages 4300–4354, 2021.


**Questions:**

As listed in weakness.

**Limitations:**

This work does not pose any negative impact.

---

> ### Author Rebuttal · Authors · 2023-08-07
>
> Thanks for your valuable comments. We will address your concerns and questions below.
>
> ---
>
> In particular, we believe there are some misunderstandings about the distinction between the non-stationary *stochastic* MDPs (studied in other works) and the non-stationary *adversarial* MDPs (this paper), as well as the lower bound argument. Those confusions could potentially be due to our lack of sufficient explanations in the paper. We would take this opportunity to make the clarifications below. If your concerns are appropriately addressed, please consider updating your score. Thanks!
>
> ---
>
> Q1: "The algorithm and result of this paper only apply to finite state case, which makes the result less significant."
>
> A1: We clarify that our algorithm can be applied to MDPs with the large state case. In Appendix B, we show that the computational complexity is independent of the number of state $S$. Additionally, our regret bound is also independent of $S$ (please kindly check the supplementary version, in which we have refined the analysis and obtained bounds independent of $S$).
>
> ---
>
> Q2: "The proof of the lower bound lacks rigor in several aspects. Firstly, the proof of the lower bound is divided into two separate cases, which is not in line with the definition of a lower bound. Ideally, the author should construct one hard instance that encompasses both cases. Secondly, in the proof of the second case (dynamic regret case), the proof merely takes the maximum between the results of the two cases with respect to the range of $\Gamma$."
>
> A2: We respectfully disagree with this comment. It’s a typical strategy to divide the entire problem into several distinct cases to establish the lower bound, which allows us to find hard instances more conveniently. This is both reasonable and sound. Let's consider a problem denoted as $A$, which consists of two cases, $A_1$ and $A_2$, where the difficulty of case $A_1$ is represented by $L_{A_1}$ and the difficulty of case $A_2$ by $L_{A_2}$. In this context, it is evident that problem $A$ cannot be easier than either case $A_1$ or case $A_2$, since all the difficult instances of cases $A_1$ and $A_2$ are encompassed within problem $A$. Consequently, the difficulty of problem $A$ is at least equal to the maximum difficulty between cases $A_1$ and $A_2$. Thus, we can assert that the difficulty of problem $A$ is $\max(L_{A_1}, L_{A_2}) \geq (L_{A_1} + L_{A_2}) / 2 \geq \Omega(L_{A_1} + L_{A_2})$. We will add more explanations in the revised version.
>
> ---
>
> Q3: "more comprehensive and detailed discussions with non-stationary RL are needed. "
>
> A3: Thank you for the suggestion. We will certainly enrich the discussion on the existing literature. However, it's crucial to emphasize that non-stationary MDPs can be divided into two main threads: (i) non-stationary *stochastic* MDPs, which most previous works are concerned with; and (ii) non-stationary *adversarial* MDPs, which this paper focuses on. The two settings, along with their respective results and algorithmic components, are very different and generally incomparable. They can be viewed as two distinct models for addressing non-stationary online learning and decision-making processes. In fact, these two settings are typically studied independently, even in simplified scenarios such as full-information and bandit online learning. For a more detailed discussion on the significant differences between these two setups, **please refer to A1 for Reviewer Xpcd**.
>
> Furthermore, we note that all the relevant works (to the best of our knowledge) on non-stationary adversarial MDPs are covered in our paper. We will add additional discussions on other related works, including those concerning non-stationary stochastic MDPs, to provide readers with a more comprehensive understanding of the literature. Thank you.
>
> ---
>
> Q4: "does the non-stationary RL discussed in [1] cover the adversarial RL when the comparators are optimal policies in this paper?"
>
> A4: No, the reduction-based framework in [1] can handle non-stationary **stochastic** MDPs, but it *cannot* be applied to non-stationary **adversarial** MDPs even the comparators are set as optimal policies.
>
> Indeed, Wei and Luo (2020) propose a generic reduction, which requires the base learner to satisfy a certain property enjoyed by typical UCB-type algorithms. When a new instance of the base algorithm surpasses this optimistic estimator, it can be inferred that the environment has undergone changes, prompting a restart of the algorithm to disregard prior information. However, this approach of constructing an optimistic estimator using a UCB-type algorithm can only be applied effectively in a **stochastic** setting. In the **adversarial** setting, where no model assumptions are made and comparators can be *arbitrary*, this approach encounters significant difficulties.
>
> This difference highlights the significant challenges inherent in handling non-stationary adversarial MDPs. In fact, **as discussed in A2 for Reviewer Xpcd**, non-stationary adversarial online learning can be sometimes much harder than non-stationary stochastic online learning, even for the multi-armed bandit setting.
>
> Thanks for your question. We will clarify this point more clearly in the revised version.
>
> ---
>
> Q5: "It is important to discuss the relationship between the findings of [1] and this paper in such cases"
>
> A5: We clarify that [1] studies the non-stationary stochastic MDPs, which is significantly different from our non-stationary adversarial MDPs as mentioned in A4. Consequently, both our problem setting and corresponding results are not directly comparable. We should have discussed more on this point, and we will revise the paper to make it clear in the next version. Thanks!

---

> > ### Comment · Reviewer_BpVE · 2023-08-15
> >
> > Thank the authors for answering all my questions. Most of my concerns have been sufficiently addressed. Hence I raise my score. Please incoporate the discussion on related work during rebuttal in final version.
> >
> > I also see the latest response of Reviewer Xpcd and find Zhong et. al, (2021) also considering nonstationary adversarial settings. However, in the A1 to Reviewer Xpcd, the authors just claim that Zhong et. al, (2021) only considers nonstationary stochastic setting. I think the authors should also clarify this.

---

> > > ### Author Response · Authors · 2023-08-15
> > >
> > > Thanks for your comment and for raising the score! We will certainly integrate the discussion on related works during the rebuttal phase into the final version.
> > >
> > > In terms of [Zhong et al., 2021], we sincerely apologize for the oversight in our initial summary regarding related works and also thank Reviewer Xpcd for highlighting this contribution. However, as mentioned in the new response to Reviewer Xpcd, [Zhong et al., 2021] mainly address the non-stationary stochastic setting, though they indeed also explore the non-stationary adversarial setting.
> > >
> > > Their contribution to the non-stationary adversarial setting essentially extends the algorithm and results in [Fei et al, 2020] to accommodate non-stationary transition kernels with linear function approximation, *without touching the essential difficulty of handling non-stationarity in the adversarial MDPs*.
> > >
> > > In contrast, we aim to **improve** the regret bound of [Fei et al. 2020] directly. To achieve this goal, we design a meta-base two-layer structure rather than the restart mechanism used by [Fei et al, 2020] and [Zhong et al., 2021] to handle the adversarial rewards. While we acknowledge that our result does not strictly surpass prior results, this paper **makes the first non-trivial step towards obtaining optimal dynamic regret in the non-stationary adversarial MDP setting** (our result is indeed optimal in certain regimes). We believe the methodology and techniques in this work will inspire subsequent research in this area to obtain the optimal regret bound for all regimes. For a more comprehensive comparison, please refer to our response to Reviewer Xpcd. We'd also like to delve into further discussions to address any remaining concerns that the reviewer may have. Thanks!

---

### Official Review · Reviewer_Xpcd · 2023-07-05

**Soundness:** 3 good
**Presentation:** 3 good
**Contribution:** 1 poor
**Rating:** 3
**Confidence:** 4

**Summary:**

This paper explores the problem of reinforcement learning in non-homogeneous MDPs with adversarial full-information reward feedback and unknown transition kernels. The authors propose a new algorithm that has advantages in dynamic regret and does not require prior knowledge as input. The algorithm achieves optimal performance in certain cases. The paper provides a theoretical foundation for reinforcement learning in non-homogeneous MDPs by studying the upper and lower bounds of dynamic regret.

**Strengths:**

This work investigates the dynamic regret in linear mixture MDPs with adversarial rewards, which is a problem of great significance and relevance. The authors make notable contributions by offering a precise upper bound on regret as well as a lower bound for this problem. To derive the upper bound, the authors introduce new techniques, including the fix-share mechanism and a multiplicative stability lemma, which hold potential value and may be of independent interest to researchers in related fields.

**Weaknesses:**

- The authors have not included a highly related work titled "Optimistic Policy Optimization is Provably Efficient in Non-stationary MDPs," which addresses the more challenging setting of linear mixture MDPs with adversarial rewards and non-stationary transitions.

- When comparing this work with both (Fei et al., 2020) and "Optimistic Policy Optimization is Provably Efficient in Non-stationary MDPs," it is worth noting that this work demonstrates improvements in terms of the parameters K and P_T. However, it introduces an additional dependency on S_T. Since S_T can be linear in K in the worst-case scenario, the resulting regret bounds may not strictly improve upon the previous findings.

- Considering the omission of the aforementioned related work and the potential limitations of the current results in terms of the additional dependency on S_T, it may be valuable for the authors to discuss these aspects in the paper. This would help readers gain a comprehensive understanding of the work's contributions and its relationship to the existing literature.

**Questions:**

- While the fix-share mechanism and multiplicative stability lemma used in this work aim to enhance the regret upper bound, it is important to assess whether these techniques strictly improve upon existing works. The authors should provide a clear analysis or comparison demonstrating how their approach surpasses or builds upon the achievements of previous methods. If the improvements are not strictly superior but rather offer refinements or alternative perspectives, they should be clearly stated to avoid potential misconceptions.

- In addition, offering further explanations on why the reduction-based framework in (Wei and Luo, 2020) is not applicable in adversarial linear mixture MDPs, even with full-information feedback, would add value to the paper. Discussing the specific challenges or characteristics of adversarial linear mixture MDPs that render the reduction-based framework ineffective or impractical will enable readers to grasp the unique complexities of the problem domain. By providing these explanations, the authors can highlight the significance of their proposed approach and its suitability for addressing the specific challenges posed by adversarial linear mixture MDPs.

---

> ### Author Rebuttal · Authors · 2023-08-07
>
> Thanks for your helpful review. We will answer your questions individually. If your concerns are properly addressed, please consider updating the score. Thanks!
>
> ---
>
> First, the reviewer may have an important misunderstanding on the two different settings: (i) non-stationary *stochastic* MDPs (other works)  and (ii) non-stationary *adversarial* MDPs (this paper). This confusion could be due to our insufficient discussions. We'd clarify the salient differences below.
>
> Q1: a related work [Zhong et al. 2021], which is more challenging
>
> A1: Thanks for bringing this paper, which also studies the non-stationarity issue in MDPs and thus related to our work. We will undoubtedly cite and discuss it in the next version. However, it’s crucial to point out that the setting of [Zhong et al., 2021] is *fundamentally different* from ours.
>
> - They study non-stationary **stochastic** MDPs, where the reward is assumed to be *stochastically* generated by parametric models with parameters continuously drifting.
> - In contrast, we study non-stationary **adversarial** MDPs, allowing rewards *adversarially* chosen. The objective is to be competitive with a sequence of time-varying compared policies.
>
> So we respectfully disagree with the comment that setup in [Zhong et al., 2021] is more challenging than ours --  they are actually *incomparable*.  Below we highlight more detailed differences. For simplicity, we consider the fixed transition kernel scenario.
>
> - In non-stationary **stochastic** MDPs, rewards are generated **stochastically** according to some parametric models that may vary over time, for example,  $r_k(s, a) = \phi(s, a) \theta_k^* +noise$, and the aim is to optimize regret against the drifting parameter $\theta_1^*,...,\theta_K^*$.
> - Contrarily, non-stationary **adversarial** MDPs  do *not* make any stochastic assumption over the rewards $r_k$. Instead, they compete with *arbitrary* feasible sequence of time-varying compared policies, which could be chosen with hindsight as the oracle comparators that best fit the underlying environments with an optimal balance between bias (due to various factors like sample randomness) and variance.
>
> Algorithmic ingredients to handle non-stationarity are also significantly different (sliding window/restart/weights for stochastic MDPs; and two-layer structures for adversarial MDPs).
>
> To summarize, the two settings and respective algorithms/results are *incomparable*. They can be viewed as two distinct models for non-stationary online learning. Actually, in some cases the adversarial setting would be even harder, see A2 for discussions.
>
> We will include those discussions in the next version to enhance the literature review. Thanks!
>
> ----
>
> Q2: "why ... framework in (Wei and Luo, 2020) is not applicable in adversarial linear mixture MDPs"
>
> A2: This framework [Wei and Luo, 2020] can handle non-stationary **stochastic** MDPs, but cannot be applied to our adversarial case. Indeed, their reduction requires the base learner to satisfy a UCB-like property. When a new instance of base algorithm surpasses this optimistic estimator, the MASTER algorithm will suspect environment suffering changes and perform the restart. However, this strategy is challenging to apply in adversarial setting, since the construction of a UCB-type base learner is difficult when there are no statistical model assumptions.
>
> Indeed, non-stationary stochastic/adversarial online learning and decision making are usually examined independently, even in simplified scenarios such as full-information and bandit online learning. Actually, in certain instances, non-stationary adversarial setting can be *more challenging* than the non-stationary stochastic one. Consider the multi-armed bandits problem.
>
> - It is extremely difficult to handle its non-stationary *adversarial* model, and actually achieving an optimal bound without knowing the non-stationarity level is provably *impossible* for adaptive adversaries [Marinov and Zimmert, 2021], and remains open for oblivious adversaries.
>
> - In contrast, it's feasible to derive an optimal strategy for a non-stationary *stochastic* setting without the non-stationarity level using the technique of [Wei and Luo, 2022].
>
> More technical discussions can be found in P3 of [Luo et al., 2022]. We will clarify it in the revised version.
>
> T. Marinov and J. Zimmert. The Pareto Frontier of Model Selection for General Contextual Bandits. NeurIPS 2021.
>
> C.-Y. Wei and H. Luo. Non-stationary Reinforcement Learning without Prior Knowledge: An Optimal Black-box Approach. COLT 2021.
>
> H. Luo, M. Zhang, P. Zhao, and Z.-H. Zhou. Corralling a Larger Band of Bandits: A Case Study on Switching Regret for Linear Bandits. COLT 2022.
>
> ---
>
> Q3: issue on switching number $S_T$
>
> A3: We agree with the comment and acknowledge that involving data-dependent quantity $S_T$ is not fully satisfactory. But we believe our contributions are still interesting enough to the community with reasons below.
>
> - Our result is more attractive than (Fei et al,. 2020) under regimes such as stationary environment where the best base-learner rarely changes, as well as piecewise-stationary environment where the environment only change certain times (then $S_T$ aligns with the change frequency).
> - Importantly, we improve the dependence on $K$ and $P_T$, matching the lower bound established in our paper.
>
> As noted in Remark 3, the dependence on $S_T$ presents a significant technical challenge, primarily because  meta regret is now a weighted regret across all states, where the weight for each state is determined by **arbitrary** comparators $\pi_1^c,...,\pi_K^c$. This is in contrast with existing literature and presents a unique challenge. We take this for future research and will emphasize more in the revised version.
>
> ---
>
> We hope above responses sufficiently address your concerns. We're happy to provide further clarifications to additional questions during the reviewer-author discussion period. Thanks!

---

> > ### Comment · Reviewer_Xpcd · 2023-08-14
> >
> > Thanks for your comprehensive response. I agree with the authors' statement that non-stationary stochastic MDPs and non-stationary adversarial MDPs are fundamentally different. However, it seems that [zhong et al. 2021] also consider the non-stationary adversarial MDPs (see Table 1 and Theorem 4.3 in [zhong et al. 21]). Given this context, where your work isn't the first to tackle this problem and doesn't strictly surpass prior outcomes, I am inclined to maintain my original score.

---

> > > ### Author Response · Authors · 2023-08-15
> > >
> > > We thank the reviewer for highlighting this another contribution made by [Zhong et al. 2021] that we missed. [Zhong et al. 2021] indeed explored non-stationary adversarial MDPs, extending the algorithm of [Fei et al. 2020] to accommodate non-stationary transition kernels with linear function approximation. Given the fact that the restart mechanism used in [Fei et al. 2020] can be slightly modified to address the non-stationary transition and that the algorithm therein can be naturally extended beyond the tabular case, the extension from [Fei et al. 2020] to [Zhong et al. 2021] is more or less straightforward, *without touching the essential difficulty of handling non-stationarity in the adversarial MDPs*. Indeed, when examining the tabular case with the fixed transition studied by [Fei et al. 2020], the algorithm and regret bound in [Zhong et al. 2020] are the **same** as that in [Fei et al. 2020].
> > >
> > > In contrast, we aim to **improve** the regret bound of [Fei et al. 2020] directly. The restart mechanism can only achieve a dynamic regret of $O(T^{⅔} P_T^{⅓})$ in both [Fei et al. 2020] and [Zhong et al. 2021], which is not optimal as demonstrated by the lower bound established by our paper. Thus, we aim to design an **optimal** algorithm to address adversarial rewards, which is very challenging. To achieve this goal, we design a meta-base two-layer structure rather than the restart mechanism to handle the adversarial rewards. Although our result does not strictly surpass prior results, **we are the first result to achieve a minimax optimal dynamic regret of $O(\sqrt{T P_T})$ under certain regimes** (for example, when $S_T$ is small). In comparison, the regret bound achieved via **the restart mechanism remains sub-optimal across all regimes**.
> > >
> > > Therefore, our paper makes the first non-trivial step towards obtaining optimal dynamic regret in the non-stationary adversarial MDP setting. We believe the methodology and techniques in this work will inspire subsequent research in this area to obtain the optimal regret bound for all regimes. We hope the reviewer can re-evaluate the contribution of this paper based on the above facts. Thanks!

---

> > > ### Author Response · Authors · 2023-08-19
> > > **Thanks for the review! Have we properly addressed your concerns?**
> > >
> > > Dear Reviewer,
> > >
> > > We sincerely appreciate your constructive feedback and are especially grateful for bringing the paper by [Zhong et al., 2021] to our attention. We will update the paper to cite [Zhong et al., 2021] and incorporate the above discussions in the next version.
> > >
> > > Given that the author-reviewer discussion period is soon coming to an end, please let us know if our response has properly addressed the concerns and potential misunderstandings. We will be happy to provide clarification if you have any further questions. Thanks!
> > >
> > >
> > > Best,
> > >
> > > Authors

---

> > > > ### Comment · Reviewer_Xpcd · 2023-08-21
> > > >
> > > > Thank you for your response. However, my main concerns (your work isn't the first to tackle this problem and doesn't strictly improve existing results) have not been fully addressed. So I decide to keep my score.

---

### Official Review · Reviewer_yhLY · 2023-07-13

**Soundness:** 4 excellent
**Presentation:** 4 excellent
**Contribution:** 3 good
**Rating:** 8
**Confidence:** 4

**Summary:**

The paper "Dynamic Regret of Adversarial Linear Mixture MDPs" studies reinforcement learning in episodic non homogeneous MDPs with adversarial rewards in full information feed back setting. The motivation for this work is that In many applications, reward functions may be picked adversarially, and changing over time. Moreover, the state and action spaces may be huge. Hence, this paper considers adversarial rewards and linear function approximation. Previous work lacks in one of these requirements. The authors analyse that their algorithm enjoys a "dynamic regret" which is close to "optimal" by deriving a lower bound, and can recover the static setting, and improves the adversarial reward regret for tabular MDPs.

**Strengths:**

A novel algorithm is proposed that achieves good dynamic regret, under non stationary adversarial regime, without the knowledge of the non-stationarity measure, which poses many challenges for the analysis.

The analysis in the paper successfully overcame these and obtained upper bound and a matching lower bound.

Static case can be exactly recovered from these results, and in case of tabular case, it strictly improves the existing bound.

**Weaknesses:**

Though out of scope for this work, I would like to know the author's perspective on the following (since they also mentioned these in the paper).

1. This work considers full information setting, bandit setting is not considered, and much more complex.

2. The dependence on S is bad in large state space regime, and on H when the non-stationary measure is large. These could not be avoided from the analysis.



**Questions:**

Please refer to above section. I appreciate if these are discussed further by the authors.

Another question is in Remark 2. How do we retrieve the static case from Theorem 1?

**Limitations:**

The authors have adequately addressed the limitations.

---

> ### Author Rebuttal · Authors · 2023-08-07
>
> We greatly appreciate the insightful and constructive comments from the reviewer. In the following, we will respond to each of the questions raised.
>
> ---
>
> Q1: “This work considers full information setting, bandit setting is not considered, and much more complex.”
>
> A1: It would indeed be very interesting to consider the bandit setting, but as far as we know, this presents significant challenges. Investigating dynamic regret with adversarial rewards and bandit feedback, even within the more simplified online learning setting (like multi-arm bandits),  is not well understood. The high-level challenge here includes how to effectively deploy the meta-base structure given the very limited bandit feedback, and how to ensure a sufficiently small meta regret to avoid ruining the overall bound. We believe that substantial novel ideas are required to develop non-trivial guarantees for the bandit setting.
>
> ---
>
> Q2: "The dependence on $S$ is bad in large state space regime, and on $H$ when the non-stationary measure is large"
>
> A2: Thank you for your question. We would like to address your concerns as follows.
>
> - The dependency on $S$ is indeed not ideal, particularly in the large state space regime. We have addressed this issue in the supplementary-material version, where we present an alternative and refined analysis for the algorithm. This analysis achieves a dynamic regret bound that is independent of $S$, but at the price of introducing another data-dependent quantity $S_T$, which represents the switching number of the best base-learner for each round and essentially captures the degree of environmental non-stationarity.
> - The dependence on $H$ in our result aligns with the most recent research, such as the works of Fei et al. [2] and Zhao et al. [58]. It is important to note that a gap persists in the dependence of $H$ even when the transition kernel is known, as indicated by Theorem 1 and Theorem 2 of Zhao et al. [58]. Addressing this issue and closing this gap is a crucial issue for future research.
>
> In summary, while our dynamic regret guarantees are already optimal in some contexts, the limitations pointed out by the reviewer indeed exist. We will take those as future work and try to improve the result further.
>
> ---
>
> Q3: "In Remark 2, how do we retrieve the static case from Theorem 1?"
>
> A3: For the static regret, the comparators are the same, that is, $\pi_1^c = \ldots = \pi_K^c = \pi^*$, i.e., $P_T=0$. Set $\gamma = 0$, the dynamic regret in Theorem 1 becomes $\text{D-Regret}(K) \leq \frac{\eta KH^3}{2} + \frac{H \log A}{\eta}$. By setting the step size as $\eta = \sqrt{\frac{\log A}{KH^2}}$, we obtain the $O(\sqrt{H^4 K \log A})$ static regret. Thanks for your question, and we will add more explanations in the revised version.

---

> > ### Comment · Reviewer_yhLY · 2023-08-18
> >
> > I thank the authors for answering my questions. Consider including the comparable works in adversarial settings, as the other reviewers pointed out. I have no further questions, and retaining my original score.

---

### Official Review · Reviewer_cWVL · 2023-07-15

**Soundness:** 3 good
**Presentation:** 4 excellent
**Contribution:** 3 good
**Rating:** 8
**Confidence:** 2

**Summary:**

This study analyzes the problem of online learning episodic inhomogeneous MDPs with _full-information_ reward functions (changed over episodes $k$ and step $h$) and the unknown transition kernel (unchanged over episodes). The MDPs are linear mixture MDPs and the performance measure is _dynamic_ regret (against _any sequence of policies_ with a non-stationarity measure of $P_T$ which controls how varied the sequence is). The proposed algorithm improves on existing upper bounds while relaxing on the requirement of prior knowledge (that of $P_T$). A lower bound is also provided which suggests optimality of the proposed algorithm on a few problem parameters.

The presentation is excellent and the proposed algorithm seems interesting. I recommend its acceptance.

**Strengths:**

1. The presentation is excellent and technically substantive. I am not an expert on the exact topic but feel comfortable following most of the discussion.
1. The proposed algorithm is interesting and uses techniques from other (recent) works (some from adjacent fields).


**Weaknesses:**

1. From the point of view of someone who is not intimately familiar with the exact topic, the problem setting might seem a bit artificial/too restrictive, e.g., full information of reward functions. But it is clear from the cited works that many features of the setup are up-to-date with what the community is actively studying. The authors also acknowledged out some of such limitations.

**Questions:**

1. Can you provide some intuition of what the switching number of best base-learner, $S_T$, measures? I found it challenging to grasp due to the presence of the expectation taken over (counterfactual) state distribution generated by $\pi^c_k$. Are there some regimes for this parameter that are interesting (and provide intuitive connections to existing studies)?

**Limitations:**

Adequate.

---

> ### Author Rebuttal · Authors · 2023-08-07
>
> Thanks for your insightful comments! Below we will address your questions.
>
> ---
>
> Q1: “Can you provide some intuition of what the switching number of best base-learner, $S_T$, measures? Are there some regimes for this parameter that are interesting (and provide intuitive connections to existing studies)?”
>
> A1: Thanks for the question. The quantity $S_T$ denotes the switching number of the best base-learner for each round, which essentially reflects the degree of environmental non-stationarity. Consider the following two examples.
>
> 1. In the stationary environment (i.e., reward function remains unchanged), $S_T$ could be relatively small as the best base-learner would seldom change.
> 2. In the piecewise-stationary environment, $S_T$ would align with the frequency of these environmental changes.
>
> In this regard, $S_T$ can be considered as an additional measure of the level of non-stationarity.
>
> But admittedly, we acknowledge that $S_T$ is a data-dependent quantity, and one may prefer to obtain bounds that rely solely on the problem-dependent quantity, such as $P_T$. As noted in Remark 3, this would introduce technical challenges that are difficult to address. We take this issue for future research and will emphasize more on this point in the revised version.
>
> ---
>
> Q2: " I found it challenging to grasp due to the presence of the expectation taken over (counterfactual) state distribution generated by $\pi_k^c$"
>
> A2: Sorry for the confusion. We define $S_T$ by taking the expectation over the state distribution generated by $\pi_k^c$ to derive a more refined bound. The expectation can be conveniently replaced by considering the maximum over all states.
>
> ---
>
> Thank you for raising these questions. We will revise our paper to highlight the points discussed above and further improve the presentation in the next version.

---

### Official Review · Reviewer_Gpzi · 2023-07-21

**Soundness:** 3 good
**Presentation:** 3 good
**Contribution:** 3 good
**Rating:** 6
**Confidence:** 2

**Summary:**

The paper studies RL in the adversarial episodic inhomogeneous MDP setting with transition kernels. It should be noted that the paper considers full-information feedback & linear mixture MDPs (both are very strong assumptions). The authors propose an algorithm drawing inspiration from policy optimization and prediction with expert advice problems (hence the full-information feedback setting). An upper bound on the dynamic regret is provided. The authors also offer an accompanying lower bound.

**Strengths:**

The paper is generally well-written and seems technically sound. An extensive literature review is provided and the authors offer comparison and discussion with closely related works. The regret bound improves upon those in existing works, and is near-optimal.

**Weaknesses:**

Among the key technical innovations listed in the introduction compared to existing works (Fei et al [2], Cai et al [34]), I feel like the extension from tabular MDPs to linear mixtures MDPs not technically challenging, given the fact that proximal policy optimization can be naturally extended beyond the tabular setting. In that case, the real contribution lies in the improvement of the regret bounds. Could the authors highlight what are the key innovations (either in the algorithms or in the analysis) that lead to this improvement?

**Questions:**

Could the authors highlight what are the key innovations (either in the algorithms or in the analysis) that lead to this improvement?

**Limitations:**

The authors discuss some limitations & future works in the conclusion.

---

> ### Author Rebuttal · Authors · 2023-08-07
>
> Thanks for your valuable comments and suggestions. We provide our response to each question below.
>
> ----
>
> Q1: "Could the authors highlight what are the key innovations (either in the algorithms or in the analysis) that lead to this improvement?"
>
> A1: The key innovations over the prior works (especially the works of Fei et al. [2] and Cai et al. [34]) for non-stationary MDPs lie in the different ways to handle the environmental non-stationarity.
>
> Cai et al. [34] focused solely on the static regret of adversarial linear mixture MDPs where a single, fixed policy served as the comparator. Their algorithm and analysis do not support dynamic regret minimization, which requires to compare with a sequence of time-varying comparators.
>
> Fei et al. [2] studied the same problem setup as ours, i.e., the dynamic regret of adversarial linear mixture MDPs. Their algorithm follows a **restarting strategy** to handle the non-stationarity. Specifically, the algorithm will periodically restart itself to discard previous information. However, this restart period requires the prior knowledge of the environmental non-stationarity $P_T$ as an algorithmic input, which is unfavorable.
>
> In contrast, our approach tackles the environmental non-stationarity via the **two-layer meta-base framework**. We first design a policy optimization algorithm which is capable of tracking time-varying compared policies (this is achieved in a non-trivial manner, and see Lemma 1 in the paper), though the optimal step size tuning still requires the prior knowledge of $P_T$. To remove this unpleasant dependence, we introduce a meta-base two-layer structure  that concurrently maintains multiple base-learners. Each base-learner is associated with a candidate step size, ensuring that the optimal step size is well approximated when considering all base learners together. Subsequently, we employ a meta-algorithm to track the optimal base-learner and thus can achieve favorable guarantee without knowing $P_T$ ahead of time.
>
> While this framework can be kind of standard in modern online learning, several new technical challenges need to be solved when applied to online MDPs. For example, the standard dynamic regret analysis relies a telescoping argument in online convex optimization, which won't work in online MDPs because regret here is defined on the expectations across changing policies. We address this issue by leveraging a fixed-share mechanism and presenting a novel multiplicative stability lemma, as detailed in Lines 203-208. Furthermore, the standard implementation of the two-layer structure requires us to update and evaluate multiple base-learners simultaneously. However, we can only deploy a single combined policy in the environment. We handle this challenge by a new analysis. More technical highlights can be found in the end of page 2 in our paper.
>
> In summary, a proper deployment of the meta-base two-layer structure is the key innovation of our proposal method to achieve the improvement.  We appreciate your feedback and will revise the paper to more clearly emphasize these discussions in the next version. Thanks!

---

> > ### Comment · Reviewer_Gpzi · 2023-08-12
> > **Thank you**
> >
> > I would like to thank the authors for their response. I have read through and agreed with the issue pointed out by Reviewer Xpcd, especially with omitted related work. Please incorporate your response to the final version of this paper.
> >
> > In general, I think the paper is in good shape. But I should admit that although I am familiar with statistical RL, I am **not** familiar with the literature on adversarial MDPs. Hence, I cannot comment on the novelty and contribution of the paper on top of the existing work.

---

> > > ### Author Response · Authors · 2023-08-15
> > >
> > > Thanks for your response. We will certainly incorporate the discussion on related works during the rebuttal phase into the final version. To summarize, our paper **makes the first non-trivial step towards obtaining optimal dynamic regret in the non-stationary adversarial MDP setting** (our result is indeed optimal in certain regimes). To achieve this goal, we design a meta-base two-layer structure rather than the restart mechanism to handle the adversarial rewards. We believe the methodology and techniques in this work will inspire subsequent research in this area to obtain the optimal regret bound for all regimes. We'd also like to delve into further discussions to address any remaining concerns that the reviewer may have. Thanks!

---

### Decision · Program_Chairs · 2023-09-21

**Decision:**

Accept (poster)

**Comment:**

The work considers the RL problem was in the adversarial episodic linear mixture MDP setting with transition kernels, and not assumptions on the reward function. Further, in terms of feedback model the authors assume full-information feedback. It is worthwhile to note that the linear mixture MDP (as oppose to general linear MDPs) and full information assumptions are quite strict. For this setting, the authors design a policy optimization algorithm, establish upper bound on its dynamic regret, and establish a lower bound for this problem.

The key issue that was raised by reviewer Xpcd is the relation to the work of Zhong et al., 2021, Optimistic Policy Optimization is Provably Efficient in Non-stationary MDPs. Zhong et al., 2021 studied the linear MDP setting with linear reward function and derived a
 upper bound for the dynamic regret of this setting. Even after a long correspondence between reviewer Xpcd and the authors they did not come to agreement about the novelty of this work. The authors highlighted their reward function is adversarial (yet with full information access) whereas in Zhong et al., 2021 the authors made an assumption on its parametric form (that its linear in the features). Another different, that I do find interesting, is the fact the authors obtained an upper bound, as oppose to the upper bound on the authors of Zhong et al., 2021 derived. That being said, the fact in this work the authors assume the full information setting, makes their setting substantially "simpler" than the one of Zhong et al., 2021.

Nevertheless, besides of Xpcd, all other reviewers were positive and in consensus about the quality of this paper. They found the results sufficiently interesting and praised the quality of writing (for example, the literature review on the topic was positively mentioned). In light of these, I currently recommend on the acceptance of this paper: it seems to me the setting studied in this work and in Zhong et al., 2021 is sufficiently different. That being said, much resemblance exist between the settings, and for this reason the authors must cite this work and elaborate on the differences. Further, since Zhong et al., 2021 already studied dynamic regret in RL setting, I believe this work does not deserve to be accepted as a spotlight paper, but only as a poster.